# DISTRIBUTION-FREE LOWER PREDICTIVE BOUNDS FOR RIGHT-CENSORED TIME-TO-EVENT DATA VIA HYBRID QUANTILE LEARNING AND DFT-ADAPTIVE CONFORMAL CALIBRATION

## ABSTRACT

Reliable uncertainty quantification for time-to-event outcomes is challenging when observations are censored and censoring depends on covariates. While conformal prediction offers a distribution-free tool, existing methods for right-censored data typically rely on fixed, global filtering rules that ignore how censoring varies across individuals. We introduce a hybrid, model-agnostic framework that combines flexible conditional quantile learning with a Data-Filtered Threshold-adaptive (DFT-adaptive) conformal calibration scheme. A base learner, instantiated as censored quantile regression forests, is trained with censoring addressed via localized Kaplan-Meier estimation; and conformity scores are calibrated nonparametrically using covariate-dependent-censoring thresholds. Our development yields marginally valid lower predictive bound that adapts to heterogeneous censoring and scales to nonlinear settings without parametric assumptions on the censoring mechanism. We provide theoretical guarantees and supporting experiments to demonstrate that the method effectively delivers adaptive, interpretable, distribution-free uncertainty quantification for censored outcomes.

## 1 INTRODUCTION

Research on time-to-event data (often referred to as survival analysis) focuses on the time until a specific event occurs, with extensive applications in many domains, including healthcare, reliability, and online platforms. A defining feature that differentiates survival analysis from standard regression in statistics and supervised learning in machine learning is *censoring*, where the event is not observed within the study window.

### 1.1 LITERATURE REVIEW

Time-to-event prediction is central to decision-making. Classical survival models (e.g., Cox proportional hazards, accelerated failure time, and semiparametric transformation models) can perform well when their structural assumptions (e.g., proportional hazards or specific parametric forms) hold (Cox, 1972; Kalbfleisch and Prentice, 2002), but these assumptions are often violated in heterogeneous and nonlinear settings (Lee et al., 2018). At the other end of the spectrum, modern machine-learning methods have matured. Conformal prediction provides model-agnostic procedures with minimal assumptions (exchangeablility of data) (e.g., Vovk et al. (2005); Barber et al. (2021; 2023)). Practical refinements, such as split conformal prediction (e.g., Lei et al. (2013); Oliveira et al. (2024); Romano et al. (2020)), conformalized quantile regression (e.g., Romano et al. (2019); Colombo and Vovk (2020)), and distributional conformal prediction (e.g., Chernozhukov et al. (2021); Izbicki et al. (2022); Vovk and Bendtsen (2018)), are scalable and adaptive to handle practical problems; weighted/local variants further address covariate shift and heterogeneity (Tibshirani et al., 2019; Gibbs and Candès, 2021; Prinster et al., 2022)).

However, conformal prediction cannot be naively applied to censored data: uncensored observations are not representative of the full cohort, breaking exchangeability. Handling such data has attracted recent interest, and a growing line of work therefore tailors conformal methods to censored

outcomes (Qin et al., 2025). Candès et al. (2023) proposed conformalized survival analysis with a global censoring cutoff and weighted split–conformal calibration to obtain finite-sample lower predictive bounds. Gui et al. (2024) introduced covariate-adaptive (data-filtered) cutoffs that calibrate only on regions with observable outcomes, improving efficiency while retaining coverage on the supported region. Moving beyond one-sided bounds, Yi et al. (2025) constructed two-sided predictive intervals under random right-censoring by recovering the upper endpoint via inverse-probability weighting, and proposed a resampling-based conformal scheme that yields one- and two-sided intervals for right-censored data. Farina et al. (2024) developed doubly robust and efficiency–oriented calibration for prediction sets via influence–function ideas, targeting improved finite–sample efficiency when either outcome or censoring models are well specified. Davidov et al. (2025) proposed a conformalized survival framework by leveraging flexible nuisance estimation to calibrate lower bounds beyond Type–I censoring. Sesia and Svetnik (2025) introduced conformal survival bands for risk screening that calibrate group–level operating characteristics under right–censoring.

On the modeling side, tree-based learners and deep architectures provide flexible, scalable estimators (Pölsterl, 2020; Prokhorenkova et al., 2018). Random forests (Breiman, 2001) and random survival forests (RSF) (Ishwaran and Kogalur, 2007) routinely deliver strong predictive accuracy, as do neural survival models such as CQRNN (Pearce et al., 2022) and DeepEH (Zhong et al., 2021). For conditional quantiles under censoring, censored quantile regression forests (cQRF, Li and Bradic (2020)) combines forest locality with censoring adjustments to estimate survival quantiles nonparametrically. Despite this natural synergy, principled integration of conformal calibration with quantile-oriented learners tailored to censoring remains limited, and methods often rely on global cutoffs or do not explicitly filter by a learned censoring horizon (Qi et al., 2024).

## 1.2 Our Contributions

We develop a hybrid, model-agnostic framework that couples a censoring-aware quantile learner with a Data-Filtered Threshold–adaptive (DFT-adaptive) conformal calibration scheme, and refer to it as DFT–cQRF. Our work differs from—and complements—existing approaches in two key ways. First, we pair a quantile-oriented base learner tailored to censoring with a covariate-adaptive censoring horizon learned nonparametrically; this contrasts with global-cutoff calibration (Candès et al., 2023) and with methods that do not explicitly filter by a learned horizon (Davidov et al., 2025; Farina et al., 2024; Sesia and Svetnik, 2025). Second, we target sharp, distribution-free lower bounds under heterogeneous censoring and analyze when the conformal correction vanishes asymptotically given a consistent quantile learner; this provides a simple pipeline that complements resampling-based or influence-function-based calibrations (Qin et al., 2025; Farina et al., 2024). Empirically, this hybridization (DFT–cQRF) improves the coverage–efficiency trade-off in settings where censoring varies strongly with covariates, while remaining model-agnostic and assumption-lean. Specifically, we make the following contributions:

- To capture individual-level heterogeneity, we estimate a covariate-dependent censoring horizon (i.e., the latest time at which outcomes remain reliably observable). This local filter removes unsupported regions with heavy censoring, which prevents overly conservative calibration while preserving validity and yielding sharper, individualized predictions.

- Instantiating the base learner with censored quantile regression forests and using localized Kaplan–Meier-based inverse-probability-of-censoring weights, we obtain distribution-free lower predictive bounds that handle heterogeneous censoring and remain simple to implement.

- We establish finite-sample marginal coverage on the observable region and show the conformal correction vanishes as the quantile learner becomes uniformly consistent, yielding asymptotically sharp bounds. Across synthetic benchmarks and an EHR application, our method ("DFT–cQRF") achieves favorable coverage–efficiency trade-offs relative to conformal survival baselines and flexible learners without distribution-free guarantees.

## 2 Methodology and Algorithm

**Preliminaries** For any subject, let $T$ denote the event time, let $X$ denote the associated $p$-dimensional covariate vector, and let $C$ denote the censoring time. Let $\delta = \mathbb{1}(T \leq C)$ de-

note indicator function, and let $Y = \min(T, C)$ denote the observed time. Let $G(t \mid x) = P(C > t \mid X = x)$ denote the conditional survivor function for the censoring time $C$ given $X = x$. For any $0 < \alpha < 1$, consider the conditional $\alpha$-quantile function of $T$ given $X = x$: $Q_\alpha(x) = \inf\{t \in \mathbb{R}^+ : P(T \leq t \mid X = x) \geq \alpha\}$. We define the covariate-dependent censoring threshold function $\tau(x)$ as: $\tau(x) = \sup\{t \in \mathbb{R}^+ : P(C \geq t \mid X = x) > 0\}$, which represents the maximal time up to which an individual with covariate profile $X = x$ may be observed (i.e., not censored). It corresponds to the right endpoint of the support of the conditional distribution of $C$ given $X = x$: If $t > \tau(x)$, then $P(C \geq t \mid X = x) = 0$. Basically, $\tau(x)$ estimates the maximum reliable observation time at covariate value x, based on the censoring times observed in the data. $\tau(x)$ tells us how far we can trust the observed $Y$ for a given covariate profile $x$, because if $Y > \tau(x)$ then it may fall in a region with high or total censoring, and any inference beyond $\tau(x)$ becomes statistically unreliable so that we cannot estimate quantiles or coverage well there. Consequently, we call $\tau(x)$ the censoring horizon function.

For $i = 1, \ldots, n$, let $(X_i, Y_i, T_i, C_i, \delta_i)$ denote random variables that are independent and identically distributed that follow the same distribution as $(X, Y, T, C, \delta)$. We use lowercase letters, $x_i$, $y_i$, $t_i$ and $c_i$, to denote realized values of the corresponding random variables. Let $\mathcal{D}_n = \{(x_i, y_i, c_i, \delta_i) : i \in \mathcal{I}\}$ denote available data, which are randomly split into a training set $\mathcal{D}_{\text{train}} \triangleq \{(x_i, y_i, c_i, \delta_i) : i \in \mathcal{I}_t\}$ and a calibration set $\mathcal{D}_{\text{cal}} \triangleq \{(x_i, y_i, c_i, \delta_i) : i \in \mathcal{I}_c\}$, where $\mathcal{I}, \mathcal{I}_t$, and $\mathcal{I}_c$ denote the index set for the data $\mathcal{D}_n, \mathcal{D}_{\text{train}}$ and $\mathcal{D}_{\text{cal}}$, respectively. That is, $\mathcal{I}_t \cup \mathcal{I}_c = \mathcal{I}$ and $\mathcal{I}_t \cap \mathcal{I}_c = \emptyset$. Let $n, n_t$ and $n_c$ denote the size of $\mathcal{I}, \mathcal{I}_t$ and $\mathcal{I}_c$, respectively.

**Methodology**  Given right-censored data, our goal is to construct an informative lower predictive bound for a future subject's event time, given covariates $X = x$, with distribution-free marginal coverage. We adopt a hybrid framework, which combines a flexible base learner estimates conditional survival quantiles and a Data-Filtered Threshold–adaptive (DFT-adaptive) conformal calibration step. Specifically, we (i) restrict prediction to the observable region defined by the estimated censoring horizon $\tau(x)$; (ii) compute conformity scores that quantify how well calibration outcomes align with the predicted quantiles under censoring; and (iii) calibrate these scores to obtain an empirical adjustment. Our procedure consists of the following five steps:

Step 1 **Estimate Covariate-Dependent Censoring Horizon:**
To gain robustness, we use the kernel regression method to estimate $\tau(x)$ by:

$$\hat{\tau}_{kern}(x) = \sup\left\{t \in \mathbb{R}^+ : \frac{\sum_{i \in \mathcal{I}} \mathbb{1}(c_i \geq t) K_h(\|x_i - x\|)}{\sum_{i \in \mathcal{I}} K_h(\|x_i - x\|)} > 0\right\} \tag{1}$$

for $x \in \mathcal{X}$ where $K_h$ is a kernel function with bandwidth $h$, and $\mathbb{1}(\cdot)$ is the indicator function.

Step 2 **Train a Base Quantile Model:**
To estimate the conditional quantile function $Q_\alpha(x)$ with right-censoring effects accounted for, we fit a censored quantile regression forest (cQRF) on $\mathcal{D}_{\text{train}}$, as detailed below:

(a) Let $B$ denote the number of trees, which is a user-specified hyperparameter, typically chosen through cross-validation or set to a default value (e.g., 100 or 500) to ensure stability of the estimated quantiles; then we fit a forest using log-rank splitting rule, a censoring-aware splitting rule.

(b) For $x \in \mathcal{X}$ and $i \in \mathcal{I}_t$, calculate the forest weights:

$$\tilde{w}_i(x) = \frac{w_i(x)}{\sum_{j \in \mathcal{I}_t} w_j(x)}, \tag{2}$$

where $w_i(x) = \frac{1}{B} \sum_{b=1}^{B} \mathbb{1}\{x \text{ and } x_i \text{ are in same leaf of tree } b\}$.

(c) We construct a localized-forest–based weights that measure the similarity between $x$ and covariates in the training data, leading to the estimator:

$$\hat{G}(t \mid x) = \prod_{j: \tilde{c}_{(j)} \leq t} \left(1 - \frac{\sum_{i \in \mathcal{I}_t} \tilde{w}_i(x) \cdot \mathbb{1}\{\tilde{c}_i = \tilde{c}_{(j)}, \delta_i = 0\}}{\sum_{i \in \mathcal{I}_t} \tilde{w}_i(x) \cdot \mathbb{1}\{\tilde{c}_i \geq \tilde{c}_{(j)}\}}\right), \tag{3}$$

where $\tilde{c}_i = \min(y_i, \hat{\tau}_{kern}(x_i))$, and $\tilde{c}_{(j)}$ denotes the $j$-th ordered distinct value among $\{\tilde{c}_i : i \in \mathcal{I}_t\}$. Then for $i \in \mathcal{I}_t$ with $\delta_i = 1$, we compute inverse probability censoring weights (IPCW):

$$\omega_i(x) = \frac{\tilde{w}_i(x)}{\hat{G}(y_i \mid x_i)}. \tag{4}$$

(d) Using the inverse probability of censoring weights (IPCW), we estimate the $\alpha$-quantile of $Q_\alpha(x)$ by the weighted empirical quantile of $\{y_i : i \in \mathcal{I}_t\}$:

$$\hat{Q}_\alpha^{\mathrm{cQRF}}(x) = \inf \left\{ t \in \mathbb{R}^+ : \sum_{i \in \mathcal{I}_t} \omega_i(x) \cdot \mathbb{1}\{y_i \le t\} \ge \alpha \right\} \tag{5}$$

for $x \in \mathcal{X}$, which can be implemented by the R package `grf` for quantile regression forests and survival forests.

**Step 3 Compute Conformity Scores:**
For $i \in \mathcal{I}_c$, define the conformity score:

$$s_i = \begin{cases} \hat{Q}_\alpha^{\mathrm{cQRF}}(x_i) - y_i & \text{if } \delta_i = 1 \text{ and } y_i \le \hat{\tau}(x_i), \\ \infty & \text{otherwise.} \end{cases}$$

Let $\mathcal{S}_{\text{finite}} = \{s_i : s_i < \infty, i \in \mathcal{I}_c\}$ denote the finite scores for subjects in the calibration set $\mathcal{I}_c$.

**Step 4 Calibrate the Quantile:**
Let $m$ denote the cardinality of $\mathcal{S}_{\text{finite}}$. Order the finite scores in $\mathcal{S}_{\text{finite}}$ as $s_{(1)} \le \cdots \le s_{(m)}$. For given $0 < \alpha < 1$, compute the empirical $(1 - \alpha)$-quantile of the conformity scores in $\mathcal{S}_{\text{finite}}$ in Step 3 by setting

$$k_\alpha = \left\lceil (1 - \alpha)(m + 1) \right\rceil \qquad \text{and} \qquad \hat{q}_{1-\alpha} := s_{(k_\alpha)} \quad (\hat{q}_{1-\alpha} = +\infty \text{ if } m = 0).$$

**Step 5 Construct the Lower Predictive Bound:**
For a new test point $X = x$, define the lower predictive bound as:

$$\hat{T}^\alpha(x) = \hat{Q}_\alpha^{\mathrm{cQRF}}(x) - \hat{q}_{1-\alpha}.$$

---

**Algorithm 1** DFT–cQRF: Distribution-Free Adaptive Conformal Lower Bound under Right Censoring

---

**Require:** Data $\mathcal{D}_n = \{(x_i, y_i, c_i, \delta_i) : i \in \mathcal{I}\}$; significance level $\alpha$.
1: Split the subject index set $\mathcal{I}$ into training set $\mathcal{I}_t$ and calibration set $\mathcal{I}_c$.
2: **for** each training point $i \in \mathcal{I}_t$ **do**
3:     (i) Estimate $\tau(x)$ using (1)
4:     (ii) Calculate $w_i(x)$ and and $\tilde{w}_i(x)$ as in (2)
5:     (iii) Estimate $G(t \mid x)$ as in (3)
6:     (iv) With $\delta_i = 1$, compute inverse probability censoring weights $\omega_i(x)$ as in (4)
7:     (v) Estimate $Q_\alpha(x)$ using $\hat{Q}_\alpha^{\mathrm{cQRF}}(x)$ in (5)
8: **end for**
9: **for** each calibration point $i \in \mathcal{I}_c$ **do**
10:     **if** $\delta_i = 1$ and $y_i \le \hat{\tau}(x_i)$ **then**
11:         Compute conformity score $s_i = \hat{Q}_\alpha^{\mathrm{cQRF}}(x_i) - y_i$
12:     **else**
13:         Set $s_i = \infty$
14:     **end if**
15: **end for**
16: Let $\mathcal{S}_{\text{finite}} = \{s_i : s_i < \infty\}$
17: Compute $\hat{q}_{1-\alpha}$ as the $(1 - \alpha)$-quantile of $\mathcal{S}_{\text{finite}}$)
18: **Return:** For any test point $x$, output the lower predictive bound $\hat{T}^\alpha(x) = \hat{Q}_\alpha^{\mathrm{cQRF}}(x) - \hat{q}_{1-\alpha}$

---

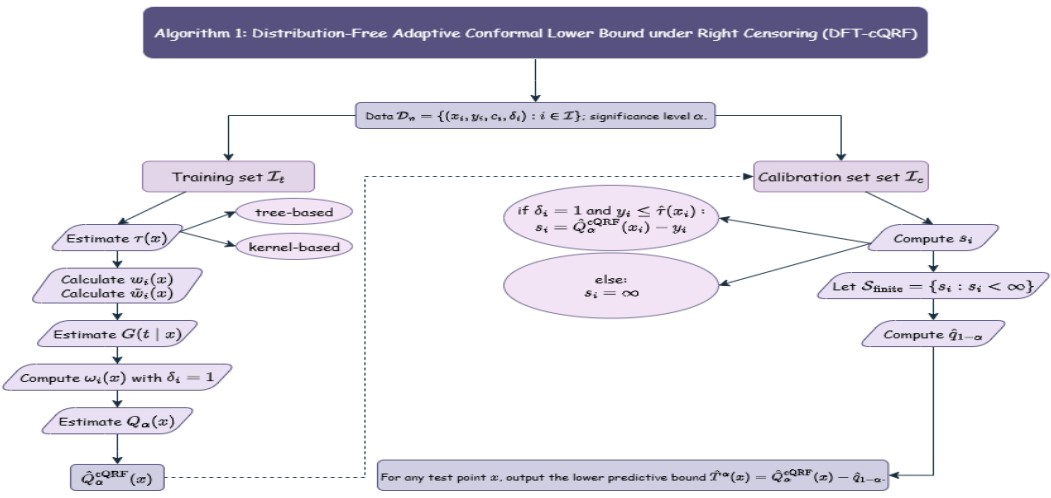

Figure 1: Pipeline diagram for the proposed DFT-cQRF method.

We introduce the DFT–cQRF procedure, which hybridizes a censoring-aware quantile learner with a data-filtered, threshold-adaptive conformal calibration scheme; pseudocode is given in Algorithm 1 and a pipeline diagram is presented in Figure 1.

An open-source implementation will be released on GitHub upon acceptance. Our method differs from prior conformal approaches in two main ways: conditional-quantile learning and covariate-adaptive calibration. First, rather than the predominantly linear quantile estimators emphasized in existing work (e.g., Gui et al. (2024)), we develop censored quantile regression forests (cQRF) that form locally weighted, censoring-adjusted empirical quantiles by integrating forest weights with inverse-probability-of-censoring weights (IPCW). This yields a model-agnostic learner that requires no parametric assumptions on the censoring mechanism and accommodates both linear and nonlinear regimes. Second, in contrast to the universal cutoff $\tau$ commonly used in censored conformal methods (e.g., Candès et al., 2023; Gui et al., 2024), we estimate a covariate-dependent censoring horizon $\hat{\tau}(x)$ via kernel nonparametric regression. Our threshold yields a local, data-adaptive filtering rule that handles censored and uncensored observations appropriately by calibrating only on those expected to be observed at their covariate values. Consequently, our method delivers sharper predictive lower bounds and improved efficiency under heterogeneous censoring, while retaining interpretability.

## 3 THEORETICAL RESULTS

This section provides theoretical guarantees for the methodology in Section 2; proofs and regularity conditions are deferred to the appendix. First, we analyze the key estimators underlying the procedure by establishing consistency for (i) the kernel-estimator $\hat{\tau}_{kern}(x)$ of the censoring horizon $\tau(x)$, (ii) the localized-forest–based weighted estimator $\hat{G}(t \mid x)$ of the censoring cumulative function $G(t \mid x)$, and (iii) the cQRF estimator $\hat{Q}_\alpha^{cQRF}(x)$ of the quantile function $Q_\alpha(x)$.

**Proposition 1.** *Assume Conditions* (K1)–(K4) *in Appendix A. Then there exist constants $c_1$ and $c_2$ depending on $(K, L, f_X)$ such that*

$$\sup_{x \in \mathcal{X}} \left| \hat{\tau}_{kern}(x) - \tau(x) \right| = O_p\left( c_1 h^\gamma + c_2 \sqrt{\frac{\log n_t}{n_t h^p}} \right),$$

*where $\gamma$ is the Hölder smoothness exponent in Condition* (K3).

Uniform consistency of the kernel estimator $\hat{\tau}_{kern}(x)$ can be established by properly choosing the bandwidth $h$. It is immediate from Proposition 1 that taking $h \asymp (\log n_t / n_t)^{1/(2\gamma + p)}$ yields

$$\sup_{x \in \mathcal{X}} \left| \hat{\tau}_{kern}(x) - \tau(x) \right| \xrightarrow{p} 0 \qquad \text{as } n_t \to \infty.$$

**Proposition 2.** *Under Conditions (C1)–(C4) in Appendix A, for any fixed $\varepsilon > 0$,*

$$\sup_{x \in \mathcal{X}} \sup_{t \leq \tau(x) - \varepsilon} \left| \hat{G}(t \mid x) - G(t \mid x) \right| \xrightarrow{p} 0.$$

The statement asserts uniform consistency of the localized-forest–based Kaplan–Meier estimator $\hat{G}(t \mid x)$ on any interior time set $[0, \tau(x) - \varepsilon]$. The $\varepsilon$-margin is essential: at $t = \tau(x)$ the risk set degenerates (with $G(t \mid x)$ close to 0), so uniform control cannot generally hold up to the boundary.

**Proposition 3.** *Assume Conditions (A1)-(A3) in Appendix A and those in Propositions 1 and 2. Then as $n_t \to \infty$,*

$$\sup_{x \in \mathcal{X}} \left| \hat{Q}_\alpha^{\mathrm{cQRF}}(x) - Q_\alpha(x) \right| \xrightarrow{p} 0.$$

The result establishes uniform consistency of the cQRF base learner: its estimated conditional $\alpha$-quantiles converge in probability to $Q_\alpha(x)$ uniformly over $x$. Practically, this means that plug-in predictions are asymptotically unbiased for $Q_\alpha(x)$ everywhere on the covariate space. It provides a reliable foundation for subsequent conformal calibration, whose theoretic analysis is provided below.

**Theorem 1.** *Assume that the regularity conditions in Proposition 3 hold. Let $\alpha$ be a constant between 0 and 1. Then as $n_t, n_c \to \infty$,*

*(a). $\hat{q}_{1-\alpha} \xrightarrow{p} 0$*

*(b). $\sup_{x \in \mathcal{X}} \left| \hat{T}^\alpha(x) - Q_\alpha(x) \right| \xrightarrow{p} 0.$*

*Remark* 1. This theorem shows that for any fixed $\alpha \in (0,1)$, when the training and calibration data sizes approach infinity, the empirical $(1-\alpha)$ quantile of the finite conformity scores satisfies $\hat{q}_{1-\alpha} \xrightarrow{p} 0$, which reflects that no positive correction is needed asymptotically on the observable region. The conformal lower predictive bound $\hat{T}^\alpha(x)$ converges uniformly to the true conditional quantile $Q_\alpha(x)$. Hence the calibrated predictor is asymptotically sharp and unbiased for $Q_\alpha(\cdot)$.

**Theorem 2.** *Assume that the conditions in Appendix A hold and that random vector $(X, T, C)$ makes the distribution of $\mathcal{D}_{train} \cup \{X, T, C\}$ and of $\{X, T, C\} \cup \mathcal{D}_{cal}$ exchangeable. Then for $\hat{T}^\alpha(x)$ obtained in Section 2 with $0 < \alpha < 1$,*

*(a) For any given $x \in \mathcal{X}$, $\hat{T}^\alpha(x)$ is non-decreasing in $\alpha$ almost surely;*

*(b) Given $\mathcal{D}_{train}$, then for a future pair $(X_{n+1}, T_{n+1})$,*

$$P\left\{ T_{n+1} \geq \hat{T}^\alpha(X_{n+1}) \,\middle|\, S_{n+1} < \infty, \, \mathcal{D}_{train} \right\} \geq 1 - \alpha.$$

*Remark* 2. The theorem states that, conditionally on the test point being observable (i.e., $S_{n+1} < \infty$), the true event time exceeds the predicted lower bound $\hat{T}^\alpha(X)$ with probability at least $1 - \alpha$, without distributional assumptions beyond exchangeability and the splitting scheme. Hence the method is finite-sample and distribution-free even under right-censoring. Moreover, the lower bound $\hat{T}^\alpha(X)$ is monotone in the miscoverage level $\alpha$: as $\alpha$ increases, the lower bound weakly increases, yielding nested, easy-to-interpret prediction sets across coverage levels.

**Theorem 3.** *Assume the conditions in Theorem 2. Given $0 < \alpha < 1$, let $c$ denote a positive constant such that*

$$P\left\{ Y_{n+1} \geq \hat{Q}_\alpha^{\mathrm{cQRF}}(X_{n+1}) - c \,\middle|\, S_{n+1} < \infty, \, \mathcal{D}_{train} \right\} \geq 1 - \alpha.$$

*Then the split-conformal choice $c = \hat{q}_{1-\alpha}$ is almost surely the smallest.*

*Remark* 3. Among all constant shifts that achieve the target coverage, the split–conformal correction $\hat{q}_{1-\alpha}$ is minimal almost surely. Consequently, the calibrated lower bound $\hat{T}^\alpha(x)$ is the *sharpest* (least conservative) distribution-free bound obtainable within the class of constant-shift adjustments: any smaller $c$ would under-cover, while any larger $c$ is unnecessarily conservative. This optimality holds for finite-sample settings, conditional on $\mathcal{D}_{\mathrm{train}}$ and on the test point being observable ($S_{n+1} < \infty$).

## 4 EXPERIMENTS

**Implementation details.** To assess the performance of DFT-cQRF (Section 2), we also consider an alternative estimator of $\tau(x)$ obtained via a regression tree in Step 1. We refer to the resulting two variants as **DFT–cQRF_k** (kernel estimator of $\tau(x)$) and **DFT–cQRF_t** (tree-based estimator of $\tau(x)$). We evaluate these variants against five benchmarks: Cox proportional hazards (**Cox**), censored quantile regression forest (**cQRF**), conformalized survival analysis (**CSA**, Candès et al. (2023)), distribution-free adaptive cutoff (**DFT-adaptive**, Gui et al. (2024)), and random survival forest (**RSF**). For visualization, we plot box plots of the lower predictive bound $\hat{T}^\alpha(x)$ from Step 5, with $\alpha = 0.1$. Each box-plot entry summarizes the replicate-level coverage over the evaluation split, computed as in Gui et al. (2024): $|\mathcal{E}|^{-1} \sum_{i \in \mathcal{E}} \mathbb{1}\big\{ y_i \geq \hat{T}^\alpha(x_i) \big\}$, where $\mathcal{E} = \big\{ i \in \mathcal{I}_c : \delta_i = 1,\ y_i \leq \hat{\tau}(x_i) \big\}$, with $\hat{\tau}(x)$ representing an estimate of $\tau(x)$.

**Real data.** We evaluate our methods on two public survival datasets. The first is an EHR-based cohort from the Kansas Health Information Network (KHIN) for suicide–risk prediction (Chen et al., 2024), comprising anonymized longitudinal covariates, timestamps, and suicide–attempt indicators from multiple health systems. The cohort exhibits right censoring due to incomplete follow-up and spans January 1, 2014–December 31, 2017. Among 3,500 patients, 3,187 individuals did not attempt suicide, yielding a censoring rate of $91.06\%$, where we define $T$ as the days to first suicide attempt (SA), and $C$ as the administrative censoring time (days from study start to end of observation). The second dataset is a machine-learning survival dataset from Kaggle containing global cancer patient records (2015–2024), included for contrast, with only $0.454\%$ censoring. Further details for both datasets and exploratory data analysis can be found in Appendix D.

Data are split into training and calibration sets using a ratio of either 75:25 or 50:50. To assess the impact of covariate inclusion on the EHR data, we analyze four specifications: (i) *age* only; (ii) *gender* only; (iii) *age + gender*; (iv) *age + gender + F333* (major depressive disorder, recurrent) + *F341* (dysthymia), and report the results in Figure 2. The red dashed line indicates the nominal $90\%$ target. Results are shown for 75:25 and 50:50 train–calibration splits; aquamarine denotes 75:25 and cyan denotes 50:50. Across all four cases, **DFT–cQRF_t** attains the highest coverage with the narrowest interquartile range and no outliers; its entire box lies above the $90\%$ line. **DFT–cQRF_k** is similarly strong, with a comparably tight box. **DFT-adaptive** centers near $90\%$. The **Cox** model is stable but below the line, whereas **cQRF**, **CSA**, and **RSF** show wider variability; **RSF** performs worst, often below $50\%$.

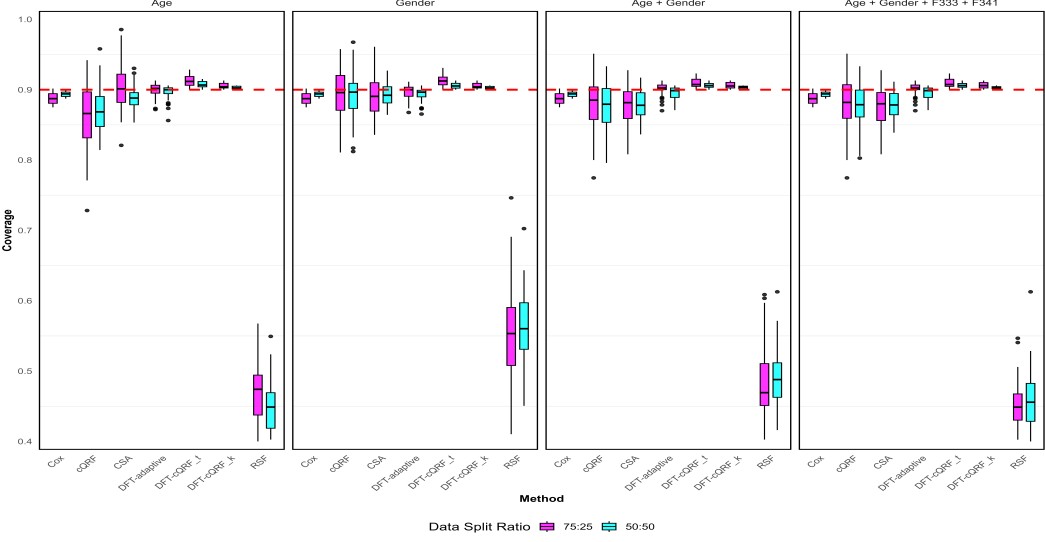

Figure 2: Analysis EHR data: empirical coverage of all methods with different covariate sets.

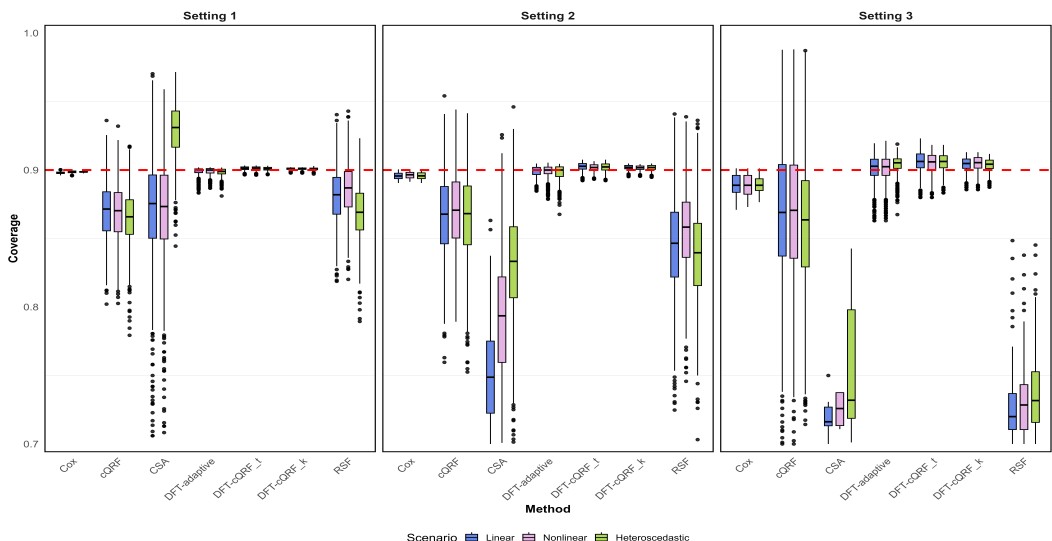

Figure 3: Synthetic experiment: empirical coverage under linear, nonlinear, and heteroscedastic relationships and different censoring proportions.

The results for the Kaggle data are presented in Figure A3 (Appendix). They show that **CSA** attains the best empirical performance, yet our methods remain competitive; in particular, **DFT–cQRF_k** is among the top performers. **CSA** outperforms here because our methods are designed to handle censored data, whereas this dataset has nearly no censoring. Even when event times are (near-)fully observed, DFT–cQRF remains efficient while retaining robustness.

**Synthetic Experiments.** We complement the real-world analysis with controlled simulations to systematically assess performance. By varying censoring rates and the covariate–outcome relationship (linear, nonlinear, heteroscedastic), we evaluate the robustness and efficiency of our methods against benchmarks used in the data analysis. We run $R = 1000$ replicates with sample size $n = 1000$ for synthetic data generated as follows. For a single covariate, $X \sim \mathcal{N}(0,1)$; for multiple covariates, $X \sim \mathcal{N}_d(0,I)$. Event times are independently generated from an accelerated failure time (AFT) model: $\log(T) = \eta(X) + \xi$, where $\xi \sim \mathrm{EV}(0,\sigma_\xi)$ (extreme value distribution), which corresponds to a Weibull baseline. We set the nominal level to $\alpha = 0.1$. Right censoring is induced independently via $C \sim \mathrm{Unif}(0,\tau)$, with $\tau$ chosen to target three censoring regimes (CR): none (Setting 1), 30% (Setting 2), and 70% (Setting 3) (tuned by pilot runs so that $\Pr(T > C) \approx$ desired rate). To probe robustness, we consider three covariate–outcome designs in the single-covariate case: (i) **Linear:** $\eta(X) = 0.5X$; (ii) **Nonlinear:** $\eta(X) = 0.5X - 0.2X^2$; and (iii) **Heteroscedastic:** $\eta(X) = 0.5X + \varepsilon$, $\varepsilon \sim \mathcal{N}(0,\sigma^2(X))$, $\sigma(X) = 0.2 + 0.3|X|$. For multiple covariates, we combined these effects across coordinates; for example, a linear–nonlinear two-covariate design is $\eta(X) = 0.5X_1 - 0.2X_2^2$. Observed data $\mathcal{D} = \{(y_i, x_i, c_i) : i \in [n]\}$ are realizations of $Y = \min(T, C)$, $X$, and $C$; we randomly split $\mathcal{D}$ into training and calibration sets in a 50:50 ratio.

We first compare methods across three covariate–outcome regimes (linear, nonlinear, heteroscedastic) and then vary censoring to assess robustness and coverage. In Figure 3, royal blue, plum, and yellow-green denote the linear, nonlinear, and heteroscedastic scenarios, respectively. Our methods (**DFT-cQRF_t**, **DFT-cQRF_k**), especially **DFT-cQRF**, consistently achieve the highest coverage with the shortest boxes (lowest variability). For **DFT-cQRF**, coverage is highest in the linear case and lower under heteroscedasticity, nevertheless, the medians and IQRs for both variantes remain above the 90% line in all scenarios. **DFT-adaptive** also performs well, with medians near or above 90% but more outliers. **Cox**, **cQRF**, **CSA**, and **RSF** generally below the line; **CSA** has the most variation across settings.

With multiple covariates (all linear, all nonlinear, or all heteroscedastic), Figure A4 (Appendix) shows results consistent with Figure 3: **DFT-cQRF_t** and **DFT-cQRF_k** remain stable with low vari-

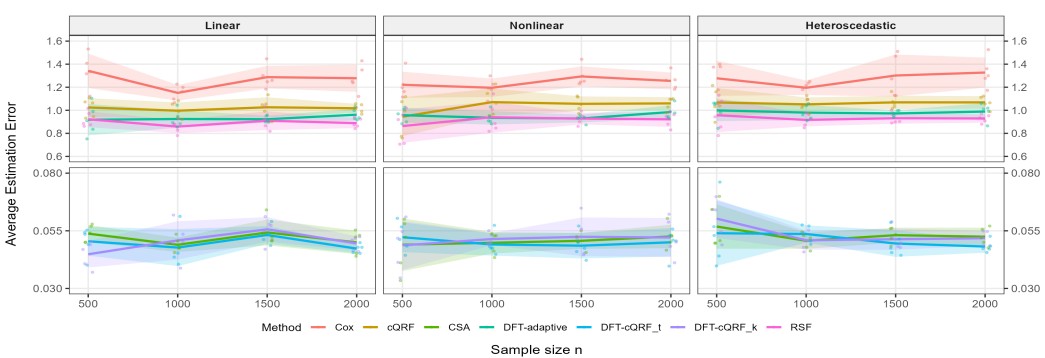

Figure 4: Synthetic experiment: average estimation error in setting 2 with CR = 0.3.

Table 1: Synthetic experiment: average estimation error with standard deviation in setting 2 with CR = 0.3.

| Scenario | $n$ | Cox | CSA | cQRF | DFT-cQRF_t | DFT-cQRF_k | DFT-adaptive | RSF |
|---|---|---|---|---|---|---|---|---|
| Linear | 500 | $1.342 \pm 0.149$ | $0.054 \pm 0.004$ | $1.024 \pm 0.080$ | $\mathbf{0.050 \pm 0.006}$ | $\mathbf{0.045 \pm 0.006}$ | $0.918 \pm 0.136$ | $0.922 \pm 0.068$ |
| | 1000 | $1.151 \pm 0.067$ | $0.049 \pm 0.003$ | $0.997 \pm 0.071$ | $\mathbf{0.048 \pm 0.008}$ | $\mathbf{0.051 \pm 0.008}$ | $0.924 \pm 0.075$ | $0.859 \pm 0.056$ |
| | 1500 | $1.288 \pm 0.098$ | $0.054 \pm 0.006$ | $1.026 \pm 0.087$ | $\mathbf{0.053 \pm 0.004}$ | $\mathbf{0.056 \pm 0.005}$ | $0.922 \pm 0.030$ | $0.909 \pm 0.081$ |
| | 2000 | $1.278 \pm 0.117$ | $0.050 \pm 0.005$ | $1.017 \pm 0.038$ | $\mathbf{0.047 \pm 0.002}$ | $\mathbf{0.050 \pm 0.003}$ | $0.962 \pm 0.050$ | $0.889 \pm 0.043$ |
| Nonlinear | 500 | $1.221 \pm 0.113$ | $0.049 \pm 0.011$ | $0.944 \pm 0.168$ | $\mathbf{0.052 \pm 0.006}$ | $\mathbf{0.049 \pm 0.011}$ | $0.958 \pm 0.064$ | $0.863 \pm 0.145$ |
| | 1000 | $1.195 \pm 0.079$ | $0.050 \pm 0.005$ | $1.071 \pm 0.117$ | $\mathbf{0.049 \pm 0.005}$ | $\mathbf{0.051 \pm 0.003}$ | $0.935 \pm 0.058$ | $0.940 \pm 0.135$ |
| | 1500 | $1.294 \pm 0.087$ | $0.051 \pm 0.003$ | $1.056 \pm 0.064$ | $\mathbf{0.049 \pm 0.006}$ | $\mathbf{0.052 \pm 0.008}$ | $0.929 \pm 0.033$ | $0.928 \pm 0.050$ |
| | 2000 | $1.255 \pm 0.073$ | $0.053 \pm 0.005$ | $1.060 \pm 0.054$ | $\mathbf{0.050 \pm 0.006}$ | $\mathbf{0.052 \pm 0.008}$ | $0.985 \pm 0.058$ | $0.921 \pm 0.065$ |
| Heteroscedastic | 500 | $1.278 \pm 0.152$ | $0.057 \pm 0.010$ | $1.068 \pm 0.124$ | $\mathbf{0.054 \pm 0.014}$ | $\mathbf{0.060 \pm 0.008}$ | $0.999 \pm 0.093$ | $0.957 \pm 0.147$ |
| | 1000 | $1.195 \pm 0.062$ | $0.051 \pm 0.003$ | $1.051 \pm 0.060$ | $\mathbf{0.054 \pm 0.004}$ | $\mathbf{0.051 \pm 0.004}$ | $0.980 \pm 0.066$ | $0.916 \pm 0.055$ |
| | 1500 | $1.302 \pm 0.182$ | $0.053 \pm 0.003$ | $1.069 \pm 0.057$ | $\mathbf{0.049 \pm 0.006}$ | $\mathbf{0.051 \pm 0.005}$ | $0.973 \pm 0.027$ | $0.931 \pm 0.048$ |
| | 2000 | $1.327 \pm 0.130$ | $0.052 \pm 0.004$ | $1.069 \pm 0.054$ | $\mathbf{0.048 \pm 0.003}$ | $\mathbf{0.052 \pm 0.003}$ | $0.989 \pm 0.074$ | $0.929 \pm 0.036$ |

ability. **cQRF** and **RSF** improve with more covariates, whereas **DFT-adaptive** declines relative to the single-covariate case. We also study mixed designs (linear + nonlinear, nonlinear + heteroscedastic, heteroscedastic + linear, and all three combined). In Figure A5 (Appendix), medium purple, dark salmon, sea green, and dark gray mark these combinations. **DFT-cQRF_t** again achieves the highest coverage; both of our methods show small between-scenario shifts, indicating robustness. **DFT-adaptive** and **Cox** are comparatively stable; **cQRF**, and especially **RSF**, vary substantially. Across Figures 3, and Figures A4 and A5 (Appendix), higher censoring produces wider boxes, longer tails, and more outliers, reflecting less information and reduced precision. Between our two variants, **DFT-cQRF_k** typically shows lower variance, whereas **DFT-cQRF_t** attains higher coverage.

Figure A7 (Appendix) summarizes the average estimation error with standard deviations (reported as mean $\pm$ sd in parentheses) across all settings and scenarios. For readability, we highlight Setting 2 in Figure 4 and Table 1. Our methods achieve comparatively low error, underscoring their competitiveness. These figures and additional tables are provided in Appendix D. Finally, we assess computational cost. Figure A6 (Appendix) reports average running times for sample sizes $n \in \{500, 1000, 1500, 2000\}$. As expected, **Cox** is consistently the fastest. **DFT–cQRF_k** requires more time due to the additional steps in our procedure; however, this overhead is reasonable given the gains in accuracy. **DFT–cQRF_t** offers a favorable accuracy–efficiency trade-off.

## 5 LIMITATIONS AND FUTURE WORK

Despite the strengths demonstrated by our theoretical analysis and experiments, several extensions remain. Our study focuses on one-sided lower bounds under right censoring with split conformal calibration and assumes exchangeability between calibration and test data. Extending the framework to two-sided prediction intervals warrants further investigation. We do not address other common features of time-to-event data, such as left truncation or interval censoring. Future work includes developing scalable conformal procedures that accommodate these settings.

## 6 ETHICS STATEMENT

We have carefully considered the ethical implications and do not anticipate any direct negative consequences arising from this work.

## 7 REPRODUCIBILITY STATEMENT

We have taken steps to ensure the reproducibility of our work. All datasets used in this study are properly referenced in the paper. To further promote transparency and facilitate future research, we will release an open-source implementation of our framework on GitHub upon acceptance.

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

APPENDICES: TECHNICAL DETAILS AND ADDITIONAL EXPERIMENTS

## A    REGULARITY CONDITIONS

Throughout, $\mathcal{X} \subset \mathbb{R}^p$ is compact. Let $f_X$ denote the probability density (or mass) function of $X$. The covariate $X$ has density $f_X$ on $\mathcal{X}$ with

$$0 < f_{\min} \leq f_X(x) \leq f_{\max} < \infty.$$

Let $f_{C|X}$ denote the conditional probability density function of $C$ given $X$, and $F_{C|X}$ denote the conditional cumulative distribution function (CDF) of the censoring time $C$ given $X$, i.e., $F_{C|X}(t \mid X = x) = 1 - G(t \mid x)$. Suppose that for $x \in \mathcal{X}$, $f_{C|X}$ has bounded support $[0, \tau(x)]$. We assume regularity conditions concerning kernel estimation, regression tree, and survival analysis that are discussed in Athey et al. (2019), van der Vaart (1998), and Kalbfleisch and Prentice (2002), which include the following conditions.

REGULARITY CONDITIONS FOR $\hat{\tau}_{kern}(x)$

(K1) **Threshhold.** $\tau(x)$ is continuous.

(K2) **Kernel & bandwidth.** $K(\cdot)$ is bounded, Lipschitz, compactly supported, $\int K(u)du = 1$; $h = h_{n_t} \downarrow 0$, and $n_t h^p / \log n_t \to \infty$ as $n_t \to \infty$.

(K3) **Hölder continuity of $G(t \mid x)$ in $x$:** There exist $\gamma \in (0, 1]$ and $L > 0$ such that

$$\sup_{t \leq \tau(x) \wedge \tau(x') - \varepsilon} |G(t \mid x) - G(t \mid x')| \leq L\|x - x'\|^\gamma \quad \text{for } x, x' \in \mathcal{X}.$$

(K4) **Positive density near the endpoint.** There exist $\eta > 0$ and $f_{C,\min} > 0$ such that

$$f_{C|X}(t \mid x) \geq f_{C,\min}$$

for all $x \in \mathcal{X}$ and $t \in (\tau(x) - \eta, \tau(x))$.

In Condition (K3), $\gamma \in (0, 1]$ is the Hölder smoothness exponent in time, and $L > 0$ is a uniform Hölder constant (independent of $x$, $s$, and $t$) that bounds the fluctuation of $G(t \mid x)$ on $[0, \tau(x)]$; when $\gamma = 1$, this reduces to a global Lipschitz condition, with $L_x$ being a Lipschitz constant.

REGULARITY CONDITIONS FOR $\hat{G}(t \mid x)$

(C1) **Uniform boundedness of censoring horizon:** $\sup_{x \in \mathcal{X}} \tau(x) \leq \tau_{\max} < \infty$ for a constant $\tau_{\max}$.

(C2) **Independence and positivity:** We assume that

    (i) Conditional independent right-censoring given $X$: $T \perp C \mid X$.

    (ii) At-risk positivity away from the tail: there exists $r_{\min} > 0$ such that

$$\inf_{x \in \mathcal{X}} \inf_{0 \leq t \leq \tau(x) - \varepsilon} P(Y \geq t \mid X = x) \geq r_{\min}.$$

(C3) **Continuity and boundedness**

    (i) For each $x$, $F_{C|X}(\cdot \mid x)$ is continuous on $[0, \tau(x)]$.

    (ii) The cumulative censoring hazard

$$\Lambda_C(t \mid x) = \int_0^t \frac{dP(Y \leq s, \delta = 0 \mid X = x)}{P(Y \geq s \mid X = x)}$$

    is uniformly bounded:

$$\sup_{x \in \mathcal{X}} \Lambda_C(\tau(x) \mid x) < \infty.$$

(C4) **Weight regularity** For each $x$, the weights $\tilde{w}_i(x) \geq 0$ satisfy:

    (i) Normalization: $\sum_{i \in \mathcal{I}_t} \tilde{w}_i(x) = 1$.

(ii) For every $\epsilon > 0$,

$$\sup_{x \in \mathcal{X}} \sum_{i \in \mathcal{I}_t} \tilde{w}_i(x) \mathbb{1}\{\|X_i - x\| > \epsilon\} \xrightarrow{p} 0.$$

(iii) As $n_t \to \infty$

$$\inf_{x \in \mathcal{X}} \left( \sum_{i \in \mathcal{I}_t} \tilde{w}_i(x)^2 \right)^{-1} \xrightarrow{p} \infty \quad \text{and} \quad \sup_{x \in \mathcal{X}} \max_{i \in \mathcal{I}_t} \tilde{w}_i(x) \xrightarrow{p} 0.$$

(iv) Conditional on $\{X_i : i \in \mathcal{I}_t\}$, the weights depend only on $\{X_i : i \in \mathcal{I}_t\}$ and are independent of $(Y_i, C_i, \delta_i) : i \in \mathcal{I}_t\}$.

ADDITIONAL CONDITIONS

(A1) **Regularity about distribution of $T$ given $X$**

Let $F_{T|X}(t|x)$ and $f_{T|X}(t \mid x)$ denote the conditional cumulative distribution function (cdf) and the density function of the event $T$ given $X = x$, respectively. For $x \in \mathcal{X}$ and $0 < \alpha < 1$,

(i) $F_{T|X}(\cdot \mid x)$ is continuous and strictly increasing near $Q_\alpha(x)$.

(ii) $f_{T|X}(\cdot \mid x)$ is bounded away from 0 in a neighborhood of $Q_\alpha(x)$.

(A2) **Continuity at the target quantile.**

(i) The conditional cdf $F^\star(t) := P(S_\alpha^*(X, T) \leq t \mid I^\star = 1)$ is continuous at its $(1 - \alpha)$-quantile $q_{1-\alpha}^\star$.

(ii) $p^* > 0$, where $S_\alpha(X, T) = Q_\alpha(X) - T$, $I^\star = \mathbb{1}\{\delta = 1, \, T \leq \tau(X)\}$, and $p^* = P(I^\star = 1)$.

(A3) **Tree / forest weights:**

(i) The partition is sufficiently fine:

$$\Delta_n := \sup_{x \in \mathcal{X}} \operatorname{diam}(\mathcal{L}(x)) \xrightarrow{p} 0,$$

where $\mathcal{L}(x)$ denotes the terminal leaf containing $x$.

(ii). Leaf mass diverges: $\inf_{x \in \mathcal{X}} \#\{i : X_i \in \mathcal{L}(x)\} \xrightarrow{p} \infty$.

# B  PROOFS OF PROPOSITIONS

**Lemma 1** (Quantile mapping). *Let $\{F_n : n = 1, 2, \ldots\}$ be cdfs and $F$ a cdf. Let $F^{-1}(u) := \inf\{t \in \mathbb{R} : F(t) \geq u\}$ denote the (left-continuous) quantile functional. For $n = 1, 2, \ldots$, define*

$$q_n = F_n^{-1}(u) \quad \text{and} \quad q = F^{-1}(u) \quad \text{for} \quad u \in (0, 1).$$

*Suppose*

*(i)* $\sup_{t \in \mathbb{R}} |F_n(t) - F(t)| \xrightarrow{p} 0$;

*(ii)* $F$ *is continuous and strictly increasing in a neighborhood of $q$.*

*Then*

$$q_n \xrightarrow{p} q.$$

*Proof.* Fix $\varepsilon > 0$. By (ii) and continuity at $q$, there exists $\eta > 0$ such that $F(q - \varepsilon) \leq u - \eta$ and $F(q + \varepsilon) \geq u + \eta$. On the event $\{\sup_t |F_n(t) - F(t)| < \eta\}$ we have $F_n(q - \varepsilon) \leq F(q - \varepsilon) + \eta \leq u$ and $F_n(q + \varepsilon) \geq F(q + \varepsilon) - \eta \geq u$. By monotonicity of $F_n$, $q - \varepsilon \leq q_n \leq q + \varepsilon$. Thus, by (i)

$$P(|q_n - q| > \varepsilon) \leq P\left( \sup_t |F_n - F| \geq \eta \right) \to 0.$$

$\square$

## A1 CONSISTENCY OF KERNEL ESTIMATOR $\hat{\tau}_{kern}(x)$

***Proof of Proposition 1.*** The proof can be carried out by modifying the existing work, as shown below. The only changes we make here are two model-specific ingredients. First, our regression target for the kernel estimator is the conditional survival function $G(t \mid x) \in [0, 1]$ rather than a mean of a real response, the Nadaraya–Watson sup-norm rates apply verbatim since the responses $\mathbb{1}(C \geq t \mid X = x)$ are bounded and the arguments are uniform in $(x, t)$ on a compact set. Second, passing from $\sup \|\hat{H} - G\|_\infty$ to an endpoint error bound uses only the standard quantile/endpoint continuity (van der Vaart's lemma) plus a local density lower bound; we require a positive conditional density near $\tau(x)$ to convert survival error into endpoint error, which is standard in quantile asymptotics. Specifically, we proceed with the following three steps.

**Step 1:**

Define the Nadaraya–Watson estimator

$$\hat{H}_x(t) = \frac{\sum_{i \in \mathcal{I}_t} \mathbb{1}\{c_i \geq t\} K_h(x_i - x)}{\sum_{i \in \mathcal{I}_t} K_h(x_i - x)}.$$

Then

$$\hat{\tau}_{kern}(x) = \sup\{t \geq 0 : \hat{H}_x(t) > 0\}.$$

Applying standard sup-norm theory for the Nadaraya–Watson estimator on compact $\mathcal{X}$ with $n_t h^p \to \infty$ yields that there exist constants $c_1$ and $c_2$ depending on $(K, L, f_X)$ such that

$$\sup_{x \in \mathcal{X}} \sup_{t \in [0, \tau(x)]} |\hat{H}_x(t) - G(t \mid x)| = O_p\big(c_1 h^\gamma + c_2 \sqrt{\log n_t / (n_t h^p)}\big) \tag{A1}$$

This is applied with the bounded "responses" $\mathbb{1}\{C \geq t\}$; the only adaptation is that the bound is used *uniformly in $(x, t)$* over the compact set $\{(x, t) : t \leq \tau(x)\}$, which those references permit. The detailed derivations are below.

**Step 1.1: Bias.**

Set

$$\hat{g}(x) := \frac{1}{n} \sum_{i=1}^n K_h(X_i - x), \quad g(x) := E\hat{g}(x) = \int K_h(u - x) f_X(u) \, du,$$

$$\hat{\mu}_t(x) := \frac{1}{n} \sum_{i=1}^n \mathbb{1}\{C_i \geq t\} K_h(X_i - x), \quad \mu_t(x) := E\hat{\mu}_t(x) = \int G(t \mid u) K_h(u - x) f_X(u) du.$$

Then $\hat{H}_x(t) = \hat{\mu}_t(x)/\hat{g}(x)$. Add–subtract $\mu_t(x)/g(x)$:

$$\hat{H}_x(t) - G(t \mid x) = \underbrace{\left[ \frac{\mu_t(x)}{g(x)} - G(t \mid x) \right]}_{\text{bias}} + \underbrace{\left[ \frac{\hat{\mu}_t(x)}{\hat{g}(x)} - \frac{\mu_t(x)}{g(x)} \right]}_{\text{stochastic+denominator}}.$$

Write

$$\mu_t(x) = \int K_h(u - x)\{G(t \mid u) - G(t \mid x)\} f_X(u) \, du + m_t(x) g(x),$$

so

$$\frac{G(t \mid x)}{g(x)} - G(t \mid x) = \frac{\int K_h(u - x)\{G(t \mid u) - G(t \mid x)\} f_X(u) \, du}{g(x)}.$$

By regularity conditions including Condition (K3) and compact support of $K$, $\|u - x\| \leq ch$ on the kernel support and

$$|G(t \mid u) - G(t \mid x)| \leq L\|u - x\|^\gamma \leq L(ch)^\gamma$$

uniformly in $t$. Since $g(x) \geq c_0 > 0$ for small $h$,

$$\sup_{x,t} \left| \frac{\mu_t(x)}{g(x)} - G(t \mid x) \right| \lesssim h^\gamma.$$

**Step 1.2: Denominator concentration.**

Standard kernel density arguments yield

$$\sup_{x \in \mathcal{X}} \left| \hat{g}(x) - g(x) \right| = O_p\left( \sqrt{\tfrac{\log n_t}{n_t h^p}} \right), \qquad \inf_{x \in \mathcal{X}} g(x) \geq c_0 > 0,$$

and hence

$$P\{\inf_x \hat{g}(x) \geq c_0/2\} \to 1 \qquad \text{as} \quad n_t \to \infty$$

**Step 1.3: Stochastic term.**

On the event $\{\inf_x \hat{g}(x) \geq c_0/2\}$,

$$\sup_{x,t} \left| \frac{\hat{\mu}_t(x)}{\hat{g}(x)} - \frac{\mu_t(x)}{g(x)} \right| \lesssim \sup_{x,t} \left| \hat{\mu}_t(x) - \mu_t(x) \right| + \sup_x \left| \hat{g}(x) - g(x) \right|.$$

It thus suffices to bound the first supremum. Consider the class

$$\mathcal{F}_h = \Big\{ (u,c) \mapsto \mathbb{1}\{c \geq t\} K_h(u - x) : x \in \mathcal{X}, \, t \in [0, \tau_{\max}] \Big\}.$$

The factor $\{\mathbb{1}\{c \geq t\} : t \in \mathbb{R}\}$ is a VC class of dimension 1, and $\{K_h(\cdot - x) : x \in \mathcal{X}\}$ has polynomial covering numbers in $1/h$ by Lipschitzness and compact support. Hence $\mathcal{F}_h$ is VC–type with entropy

$$\log N(\epsilon, \mathcal{F}_h, \|\cdot\|_\infty) \lesssim p \log(1/h) + \log(1/\epsilon).$$

Using symmetrization and Bernstein/Dudley bounds for uniformly bounded VC–type classes, together with

$$\mathrm{var}(\mathbb{1}\{C \geq t\} K_h(X - x)) \lesssim E K_h^2(X - x) \asymp h^{-p},$$

we obtain

$$\sup_{x,t} \left| \hat{\mu}_t(x) - \mu_t(x) \right| = O_p\left( \sqrt{\tfrac{\log n_t}{n_t h^p}} \right),$$

where the $\log n_t$ term subsumes $p \log(1/h)$ under Condition (K2).

**Step 1.4: Combine.**

Collecting bounds gives

$$\sup_{x \in \mathcal{X}} \sup_{t \in [0, \tau_{\max}]} \left| \hat{H}_x(t) - G(t \mid x) \right| \leq \sup_{x,t} \left| \frac{\mu_t(x)}{g(x)} - G(t \mid x) \right| + \sup_{x,t} \left| \frac{\hat{\mu}_t(x)}{\hat{g}(x)} - \frac{G(t \mid x)}{g(x)} \right|$$

$$= O_p\left( h^\gamma + \sqrt{\tfrac{\log n_t}{n_t h^p}} \right).$$

**Step 2:**

By Condition (K4), for small $u > 0$,

$$G(\tau(x) - u \mid x) \geq f_{C,\min} u. \tag{A2}$$

Then taking $u_t = 2\varepsilon_{n_t}/f_{C,\min}$, with $\varepsilon_{n_t} = c_1 h^\gamma + c_2 \sqrt{\log n_t/(n_t h^p)}$, (A2) gives

$$G(\tau(x) - u_t \mid x) \geq \varepsilon_{n_t} \qquad \text{uniformly in} \quad x.$$

Combining with (A1) yields

$$\hat{H}_x(\tau(x) - u_t) \geq G(\tau(x) - u_t \mid x) - \varepsilon_{n_t} \geq 0 \qquad \text{with high probability,}$$

hence

$$\hat{\tau}_{kern}(x) \geq \tau(x) - u_t \quad \text{uniformly in } x. \tag{A3}$$

For the upper bound, note that for any fixed $x$ and any $\eta^* \in (0, \eta)$, and for all $t \geq \tau(x) + \eta$ we have

$$G(t \mid x) = 0.$$

Because $K(\cdot)$ has compact support and $h \to 0$ by Condition (K2), the numerator $\sum_{i \in \mathcal{I}_t} \mathbb{1}\{c_i \geq t\} K((x_i - x)/h)$ in $\hat{H}_x(t)$ is 0 with high probability while the denominator stays positive, so

$$\hat{H}_x(t) = 0 \qquad \text{for} \qquad t \geq \tau(x) + \eta^*$$

and thus

$$\hat{\tau}_{kern}(x) \leq \tau(x) + \eta^*.$$

Letting $\eta^* \downarrow 0$ gives $\hat{\tau}_{kern}(x) \leq \tau(x)$ with high probability.

Finally, mapping sup-norm survival error into an endpoint error bound is a direct application of Lemma 1 using the density lower bound from Condition (K4).

$\square$

## A2 Consistency $\hat{G}(t \mid x)$

To show the Proposition 2, we first establish two lemmas. We write the weighted at-risk and censoring subdistribution processes:

$$\hat{R}_{n_t}(t \mid x) := \sum_{i \in \mathcal{I}_t} \tilde{w}_i(x) \mathbb{1}\{Y_i \geq t\} \qquad \text{and} \qquad \hat{A}_{n_t}(t \mid x) := \sum_{i \in \mathcal{I}_t} \tilde{w}_i(x) \mathbb{1}\{Y_i \leq t, \ \delta_i = 0\},$$

together with their population counterparts:

$$R(t \mid x) := P(Y \geq t \mid X = x) \qquad \text{and} \qquad A(t \mid x) := P(Y \leq t, \delta = 0 \mid X = x).$$

**Lemma 2.** *Assume conditions in Proposition 2. Then for any fixed $\varepsilon > 0$,*

$$\sup_{x \in \mathcal{X}} \sup_{t \leq \tau(x) - \varepsilon} \left| \hat{R}_{n_t}(t \mid x) - R(t \mid x) \right| \xrightarrow{p} 0,$$

$$\sup_{x \in \mathcal{X}} \sup_{t \leq \tau(x) - \varepsilon} \left| \hat{A}_{n_t}(t \mid x) - A(t \mid x) \right| \xrightarrow{p} 0$$

*as $n_t \to \infty$.*

*Proof.* Fix $x$. Conditional on $\{X_i : i \in \mathcal{T}_t\}$ and the weights, the summands are bounded and independent with means

$$\mathbb{E}[\mathbb{1}\{Y_i \geq t\} \mid X_i] = P(Y \geq t \mid X_i)$$

and

$$\mathbb{E}[\mathbb{1}\{Y_i \leq t, \delta_i = 0\} \mid X_i] = P(Y \leq t, \delta = 0 \mid X_i),$$

by independent censoring given $X$ in Condition (C2)(a) and the definition of $Y$ and $\delta$. A weighted LLN then yields

$$\hat{R}_{n_t}(t \mid x) - \sum_{i \in \mathcal{I}_t} \tilde{w}_i(x) R(t \mid X_i) \xrightarrow{p} 0 \qquad \text{uniformly in} \qquad t \leq \tau(x) - \varepsilon,$$

because the effective size $\left( \sum_{i \in \mathcal{I}_t} \tilde{w}_i(x)^2 \right)^{-1} \xrightarrow{p} \infty$ and $\max_{i \in \mathcal{I}_t} \tilde{w}_i(x) \xrightarrow{p} 0$ (Condition (C4)(iii)), together with honesty/stability (Condition (C4)(iv)). A similar result holds for $\hat{A}_{n_t}$.

By localization (Condition (C4)(ii)) and continuity in $x$ of the maps $x \mapsto R(t \mid x)$, $x \mapsto A(t \mid x)$ (which follow from dominated convergence on the compact $\mathcal{X}$, we have

$$\sum_{i \in \mathcal{I}_t} \tilde{w}_i(x) R(t \mid X_i) \xrightarrow{p} R(t \mid x) \qquad \text{uniformly on} \qquad \{(t, x) : t \leq \tau(x) - \varepsilon, \ x \in \mathcal{X}\}$$

and similarly for $A(t \mid x)$. A covering argument on the compact set yields the claimed suprema. $\square$

**Lemma 3.** *Assume conditions in Proposition 2. Then for any sequences $(A_{n_t}, R_{n_t})$ satisfying*

$$\sup_{x \in \mathcal{X}} \sup_{t \leq \tau(x) - \varepsilon} \left\| A_{n_t}(t \mid x) - A(t \mid x) \right\| \to 0$$

*and*

$$\sup_{x \in \mathcal{X}} \sup_{t \leq \tau(x) - \varepsilon} \left\| R_{n_t}(t \mid x) - R(t \mid x) \right\| \to 0,$$

*we have*

$$\sup_{x \in \mathcal{X}} \sup_{t \leq \tau(x) - \varepsilon} \left| \prod_{(0,t]} \left( 1 - \frac{dA_{n_t}(s \mid x)}{R_n(s \mid x)} \right) - \prod_{(0,t]} \left( 1 - \frac{dA(s \mid x)}{R(s \mid x)} \right) \right| \longrightarrow 0 \qquad as \quad n_t \to \infty.$$

*Proof.* On $\{t \leq \tau(x) - \varepsilon\}$, Condition (C2) gives

$$R(t \mid x) \geq r_{\min} > 0,$$

hence for $n_t$ large also

$$R_{n_t} \geq r_{\min}/2.$$

Write the logs and use the bounded variation of $A(t \mid x)$ (Condition (C3)) to control the remainder:

$$\log \prod_{(0,t]} \left( 1 - \frac{dA_{n_t}(s \mid x)}{R_{n_t}(s \mid x)} \right) - \log \prod_{(0,t]} \left( 1 - \frac{dA(s \mid x)}{R} \right)$$

$$= \int_{(0,t]} \left[ \log \left( 1 - \frac{dA_{n_t}(s \mid x)}{R_{n_t}(s \mid x)} \right) - \log \left( 1 - \frac{dA(s \mid x)}{R(s \mid x)} \right) \right].$$

Using

$$\log(1 - u) = -u - \frac{u^2}{2}\theta(u)$$

with $\theta(u)$ bounded on $[0, 1 - r_{\min}/2)$, one obtains a bound by the total variation of

$$\left| \frac{dA_{n_t}(s \mid x)}{R_{n_t}(s \mid x)} - \frac{dA(s \mid x)}{R(s \mid x)} \right|.$$

Uniform convergence of $A_{n_t}(s \mid x) \to A(s \mid x)$ and $R_{n_t}(s \mid x) \to R(s \mid x)$ then implies the desired uniform convergence of the product integrals. $\square$

***Proof of Proposition 2.*** By definition of the product-integral, the (population) censoring survival satisfies

$$G(t \mid x) = \prod_{(0,t]} \left\{ 1 - \frac{dA(t \mid x)}{R(t \mid x)} \right\},$$

and the weighted Kaplan–Meier (KM) estimator is

$$\hat{G}(t \mid x) = \prod_{(0,t]} \left\{ 1 - \frac{d\hat{A}_{n_t}(t \mid x)}{\hat{R}_{n_t}(t \mid x)} \right\}.$$

Apply Lemma 2 to get uniform convergence of the weighted processes to $(A, R)$ on the interior strip $\{t \leq \tau(x) - \varepsilon\}$. Then invoke Lemma 3 to conclude that

$$\sup_{x \in \mathcal{X}} \sup_{t \leq \tau(x) - \varepsilon} \left| \hat{G}(t \mid x) - G(t \mid x) \right| \xrightarrow{p} 0.$$

$\square$

## A3    Uniform consistency of $\hat{Q}_\alpha^{\text{cQRF}}$

***Proof of Proposition 3.*** For $t \leq \hat{\tau}_{kern}(x)$, define the localized IPCW cdf

$$\hat{F}_T(t \mid x) := \sum_{i \in \mathcal{I}_t} \tilde{w}_i(x) \frac{\delta_i \mathbb{1}\{Y_i \leq t\}}{\hat{G}(Y_i \mid X_i)},$$

where $\sum_{i \in \mathcal{I}_t} \tilde{w}_i(x) = 1$.

By (A3),

$$\hat{\tau}_{kern}(x) \geq \tau(x) - u_t \quad \text{uniformly in } x,$$

so there exists $\epsilon > 0$ such that

$$\hat{\tau}_{kern}(x) \geq \tau(x) - \epsilon > 0 \quad \text{with high probability (w.h.p.).} \tag{A4}$$

First, consider the case with known $\tau(x)$. By the IPCW identity (Stute, 1993; Akritas, 1994), a weighted LLN for forest weights (honesty/localization) (Athey et al., 2019; Wager and Athey, 2018) and uniform consistency of $\hat{G}(t \mid x)$ on $[0, \tau(x) - \varepsilon]$ in Proposition 2, the uniform convergence of $\hat{F}_T(\cdot \mid x)$ to $F_T(\cdot \mid x)$ on $[0, \tau(x) - \varepsilon]$ can be established:

$$\sup_{x \in \mathcal{X}} \sup_{t \leq \tau(x) - \varepsilon} \left| \hat{F}_T(t \mid x) - F_T(t \mid x) \right| \xrightarrow{p} 0 \tag{A5}$$

on $[0, \tau(x) - \varepsilon]$.

When $\tau(x)$ is replaced by $\hat{\tau}_{kern}(x)$, the only extra task is to guarantee that for all large $n_t$,

$$t \leq \tau(x) - \varepsilon \implies t \leq \hat{\tau}(x) \text{ for all } x,$$

so that the same uniform arguments apply on $[0, \tau(x) - \varepsilon]$. This is indeed ensured by (A4).

By Condition (A1), the inverse map $F_T \mapsto F_T^{-1}(\alpha)$ is uniformly continuous near $Q_\alpha(x)$, hence applying Lemma 1 gives

$$\sup_x \left| \hat{Q}_\alpha^{\text{cQRF}}(x) - Q_\alpha(x) \right| \xrightarrow{p} 0.$$

since $Q_\alpha(x) \leq \tau(x)$ and $\varepsilon > 0$ is arbitrary.

$\square$

## C    Proofs of Theorems

To prove Theorem 1, we first present several lemmas. For $i \in \mathcal{I}_c$, let

$$S_i = \hat{Q}_\alpha^{\text{cQRF}}(X_i) - Y_i \quad \text{and} \quad S_i^\star = Q_\alpha(X_i) - T_i,$$

and define

$$I_i = \mathbb{1}\{\delta_i = 1, Y_i \leq \hat{\tau}(X_i)\} \quad \text{and} \quad I_i^\star = \mathbb{1}\{\delta_i = 1, T_i \leq \tau(X_i)\}.$$

**Lemma 4.** *Assume regularity conditions in Proposition 1. Then*

$$\frac{1}{n_c} \sum_{i \in \mathcal{I}_c} |I_i - I_i^\star| \xrightarrow{p} 0 \quad \text{and} \quad \frac{1}{n_c} \sum_{i \in \mathcal{I}_c} I_i \xrightarrow{p} p^*.$$

*Proof.* When $\delta_i = 1$ we have $Y_i = T_i$, so

$$I_i = \mathbb{1}\{T_i \leq \hat{\tau}(X_i)\} \quad \text{and} \quad I_i^\star = \mathbb{1}\{T_i \leq \tau(X_i)\} \quad on \quad \{\delta_i = 1\}.$$

By Proposition 1,

$$\sup_x |\hat{\tau}_{kern}(x) - \tau(x)| \xrightarrow{p} 0,$$

and the indicator difference $|I_i - I_i^\star|$ can be nonzero only when $T_i$ lies in the band $(\tau(x_i) - \eta_n, \tau(x_i) + \eta_n]$ with $\eta_n := \sup_x |\hat{\tau}_{kern}(x) - \tau(x)|$. By continuity of the conditional law of $T$ given $X$ and boundedness, the probability for $T$ to fall in a vanishing band goes to zero uniformly in $x$.

An application of the weak law of large numbers yields the first claim. The second follows by the LLN since $I_i^\star$ are i.i.d. with strictly positive mean and $I_i - I_i^\star$ is negligible in average. $\square$

**Lemma 5** (Uniform score consistency on supported, uncensored points). *Assume regularity conditions in Propositions 1 and 3. Then as $n_t \to \infty$,*

$$\sup_{i \in \mathcal{I}_c : I_i^\star = 1} \left| \hat{Q}_\alpha^{\mathrm{cQRF}}(X_i) - Q_\alpha(X_i) \right| \xrightarrow{p} 0.$$

*Consequently,*

$$\sup_{i \in \mathcal{I}_c : I_i^\star = 1} |S_i - S_i^\star| \xrightarrow{p} 0.$$

*Proof.* By Proposition 3,

$$\sup_{x \in \mathcal{X}} |\hat{Q}_\alpha^{\mathrm{cQRF}}(x) - Q_\alpha(x)| \xrightarrow{p} 0.$$

If $I_i^\star = 1$ then $\delta_i = 1$ and $Y_i = T_i$, so $S_i - S_i^\star = \hat{Q}_\alpha^{\mathrm{cQRF}}(X_i) - Q_\alpha(X_i)$ and the claim follows. $\square$

**Lemma 6.** *Let $m = \sum_{i \in \mathcal{I}_c} I_i$ and define the empirical cdf based on the* finite *conformity scores*

$$F_m(t) := \frac{1}{m} \sum_{i \in \mathcal{I}_c} I_i \mathbb{1}\{S_i \le t\}, \qquad t \in \mathbb{R},$$

*with $F_m \equiv 0$ if $m = 0$. Assume regularity conditions in Propositions 1 and 3. Then as $n_t \to \infty$,*

$$\sup_{t \in \mathbb{R}} |F_m(t) - F^\star(t)| \xrightarrow{p} 0.$$

*Proof.* Write

$$|F_m(t) - F^\star(t)| = \left| \frac{1}{m} \sum_{i \in \mathcal{I}_c} I_i \mathbb{1}\{S_i \le t\} - F^\star(t) \right|$$

$$\le \underbrace{\left| \frac{1}{m} \sum I_i \left( \mathbb{1}\{S_i \le t\} - \mathbb{1}\{S_i^\star \le t\} \right) \right|}_{(A)} + \underbrace{\left| \frac{1}{m} \sum (I_i - I_i^\star) \mathbb{1}\{S_i^\star \le t\} \right|}_{(B)}$$

$$+ \underbrace{\left| \frac{1}{m} \sum I_i^\star \mathbb{1}\{S_i^\star \le t\} - \frac{1}{n_c} \sum I_i^\star \mathbb{1}\{S_i^\star \le t\} \right|}_{(C)}$$

$$+ \underbrace{\left| \frac{1}{n_c} \sum I_i^\star \mathbb{1}\{S_i^\star \le t\} - p^\star F^\star(t) \right|}_{(D)} + \underbrace{|p^\star F^\star(t) - F^\star(t)|}_{(E)}.$$

Term (E) equals $|p^\star - 1| \, F^\star(t)$ but we interpret $F_n$ as a cdf conditioned on $I^\star = 1$, so we normalize by $m$; using Lemma 4, $m/n_c \to p^\star$, thus (E) is absorbed by (C).

For (A), by Lemma 5, $\sup_{i:I_i^\star=1} |S_i - S_i^\star| \xrightarrow{p} 0$; the difference of indicators vanishes eventually except possibly when $S_i^\star$ lies within an $o_p(1)$-band around $t$, whose contribution is negligible uniformly in $t$ because $F^\star$ is cadlag and bounded.

For (B), use $\frac{1}{n_c} \sum |I_i - I_i^\star| \to 0$ and $m/n_c \to p^\star > 0$ to get $(B) = o_p(1)$ uniformly in $t$.

For (C)–(D), apply LLN with random normalization: by Lemma 4, $m/n_c \to p^\star$ and the Glivenko–Cantelli theorem for the class $\{\mathbb{1}\{S^\star \le t\} : t \in \mathbb{R}\}$ gives uniform convergence of the unnormalized averages; Slutsky's theorem yields uniform convergence after dividing by $m$. Combining the bounds and taking the supremum in $t$ gives the claim. $\square$

***Proof of Theorem 1.*** (a). By Proposition 3, for $x \in \mathcal{X}$, the estimator $\hat{Q}_\alpha^{\mathrm{cQRF}}(x)$ is consistent:

$$\hat{Q}_\alpha^{\mathrm{cQRF}}(x) \xrightarrow{p} Q_\alpha(x) \quad \text{as} \quad n_t \to \infty.$$

To evaluate the statistical behavior of conformity scores in Step 3 in Section 2, we first consider their oracle version

$$S_\alpha^*(X, T) = Q_\alpha(X) - T,$$

defined in Condition (A3), and let $q_{1-\alpha}^*$ denote the $(1 - \alpha)$-quantile of the conditional-on-observability distribution $\mathcal{L}^* \equiv \mathcal{L}(S_\alpha^*(X, T) \mid \delta = 1, \, T \leq \tau(X))$.

From Step 3 and Step 4 in Section 2, if $Q_\alpha(x)$ were known, then

$$P(S_\alpha^*(X, T) \leq 0 \mid X = x) = P(T \geq Q_\alpha(x) \mid X = x) = 1 - \alpha, \tag{A6}$$

where the last steps is due to the definition of $Q_\alpha(x)$. Therefore, $0$ is the unique $(1 - \alpha)$-quantile of the conditional-on-observability distribution $\mathcal{L}^*$, namely,

$$q_{1-\alpha}^* = 0.$$

Then, applying Lemma 6 and Proposition 1 leads to

$$\hat{q}_{1-\alpha} \xrightarrow{p} 0, \quad \text{as } n_c \to \infty.$$

(b). Applying Proposition 3 with $\hat{q}_{1-\alpha} \xrightarrow{p} 0$, we obtain

$$\hat{T}^\alpha(x) = \hat{Q}_\alpha^{\text{cQRF}}(x) - \hat{q}_{1-\alpha} \xrightarrow{p} Q_\alpha(x) \qquad \text{as } n_t, n_c \to \infty.$$

$\square$

***Proof of Theorem 2.*** (a) For any $x \in \mathcal{X}$ and $0 < \alpha_1 < \alpha_2 < 1$, by the construction of $\hat{Q}_\alpha^{\text{cQRF}}(x)$ in Step 2 in Section 2,

$$\hat{Q}_{\alpha_1}^{\text{cQRF}}(x) \leq \hat{Q}_{\alpha_2}^{\text{cQRF}}(x) \quad a.s.$$

Clearly, by definition,

$$\hat{q}_{1-\alpha_1} \geq \hat{q}_{1-\alpha_2} \quad a.s.$$

Combining both, we obtain:

$$\hat{T}^{\alpha_1}(x) = \hat{Q}_{\alpha_1}^{\text{cQRF}}(x) - \hat{q}_{1-\alpha_1} \leq \hat{Q}_{\alpha_2}^{\text{cQRF}}(x) - \hat{q}_{1-\alpha_2} = \hat{T}^{\alpha_2}(x) \quad a.s.$$

(b) Let $m$ denote the cardinality of $\mathcal{S}_{\text{finite}}$. Sort them in ascending order as

$$s_{(1)} \leq s_{(2)} \leq \cdots \leq s_{(m)}.$$

For given $0 < \alpha < 1$, compute the empirical $(1 - \alpha)$-quantile of the conformity scores in $\mathcal{S}_{\text{finite}}$ in Step 3 in Section 2 by setting

$$k_\alpha = \left\lceil (1 - \alpha)(m + 1) \right\rceil$$

and

$$\hat{q}_{1-\alpha} = \begin{cases} s_{(k_\alpha)}, & \text{if } m > 0 \\ +\infty, & \text{if } m = 0. \end{cases}$$

For a new point $(X, T, C)$ with censoring indicator $\delta$, calculate the conformity score:

$$S = \begin{cases} \hat{Q}_\alpha^{\text{cQRF}}(X) - Y, & \text{if } \delta = 1 \text{ and } Y \leq \hat{\tau}(X), \\ \infty, & \text{otherwise.} \end{cases} \tag{A7}$$

Observe that, for any $c \geq 0$,

$$\{ Y_{n+1} \geq \hat{Q}_\alpha^{\text{cQRF}}(X_{n+1}) - c \} \quad \Longleftrightarrow \quad \{ S_{n+1} \leq c \}.$$

By sample splitting, conditional on $\mathcal{D}_{\text{train}}$ the multiset

$$\{ S_i : i \in \mathcal{S}_{\text{finite}} \} \cup \{ S_{n+1} \}$$

is exchangeable on the event $\{S_{n+1} < \infty\}$. Hence the rank

$$R := 1 + \#\{i \in \mathcal{S}_{\text{finite}} : S_i \leq S_{n+1}\}$$

is uniformly distributed on $\{1, \ldots, m+1\}$ conditional on $\{S_{n+1} < \infty, \mathcal{D}_{\text{train}}\}$. Therefore,

$$P\Big\{ S_{n+1} \leq S_{(k_\alpha)} \,\Big|\, S_{n+1} < \infty, \mathcal{D}_{\text{train}} \Big\} = P\Big\{ R \leq k_\alpha \,\Big|\, S_{n+1} < \infty, \mathcal{D}_{\text{train}} \Big\}$$

$$= \frac{k_\alpha}{m+1} \geq 1 - \alpha,$$

where the last inequality uses the definition of $k_\alpha = \lceil (1-\alpha)(m+1) \rceil$. Equivalently,

$$P\Big\{ Y_{n+1} \geq \hat{Q}_\alpha^{\text{cQRF}}(X_{n+1}) - \hat{q}_{1-\alpha} \,\Big|\, S_{n+1} < \infty, \mathcal{D}_{\text{train}} \Big\} \geq 1 - \alpha.$$

$\square$

***Proof of Theorem 3.*** Split the available data into a proper training set (used to fit the predictor) and a calibration set. Fit the conditional $\alpha$-quantile predictor $\hat{Q}_\alpha^{\text{cQRF}}(\cdot)$ on the proper training set only, so that, conditional on $\mathcal{D}_{\text{train}}$ (which includes the fitted predictor and the calibration covariates), the calibration responses and the test point remain exchangeable.

Suppose $c \geq 0$ satisfies

$$P\Big\{ Y_{n+1} \geq \hat{Q}_\alpha^{\text{cQRF}}(X_{n+1}) - c \,\Big|\, S_{n+1} < \infty, \mathcal{D}_{\text{train}} \Big\} \geq 1 - \alpha,$$

i.e.,

$$P\Big\{ S_{n+1} \leq c \,\Big|\, S_{n+1} < \infty, \mathcal{D}_{\text{train}} \Big\} \geq 1 - \alpha.$$

If $c < S_{(k_\alpha)}$, then necessarily

$$\{S_{n+1} \leq c\} \subseteq \{S_{n+1} < S_{(k_\alpha)}\},$$

and exchangeability (uniform rank) yields

$$P\Big\{ S_{n+1} \leq c \,\Big|\, S_{n+1} < \infty, \mathcal{D}_{\text{train}} \Big\} \leq P\Big\{ S_{n+1} < S_{(k_\alpha)} \,\Big|\, S_{n+1} < \infty, \mathcal{D}_{\text{train}} \Big\}$$

$$= \frac{k-1}{m+1} < 1 - \alpha,$$

a contradiction. Hence any $c$ that attains the desired conditional coverage must satisfy

$$c \geq S_{(k_\alpha)} = \hat{q}_{1-\alpha} \quad \text{a.s.}$$

Since we have already shown that $c = \hat{q}_{1-\alpha}$ achieves coverage, it follows that $\hat{q}_{1-\alpha}$ is a.s. the smallest such constant.

**Tie remark.** If the conditional distribution of $S_{n+1}$ (given $\mathcal{D}_{\text{train}}$) is continuous, then ties among $\{S_i : i \in \mathcal{S}_{\text{finite}}\}$ occur with probability zero and the smallest constant is unique. With ties, the coverage function $c \mapsto P\{S_{n+1} \leq c \mid S_{n+1} < \infty, \mathcal{D}_{\text{train}}\}$ is right-continuous and jumps only at $\{S_{(j)} : j = 1, \ldots, m\}$, so the leftmost $c$ achieving coverage is still $S_{(k_\alpha)}$. $\square$

## D ADDITIONAL EXPERIMENTS

### EXPLORATORY DATA ANALYSIS

**EHR data.** The EHR cohort includes 3,500 patients and 116 features: a binary suicide-attempt (SA) outcome, demographics (e.g., gender, age), and longitudinal ICD codes. Figure A1 shows age distributions among SA cases by gender. Overall patterns are similar across sexes, though among those with SA, females exhibit a wider age range and a higher median age. We define $T$ as days to SA, with $\delta = 1$ if SA occurs during follow-up and 0 otherwise (censored). In total, 3,187 individuals

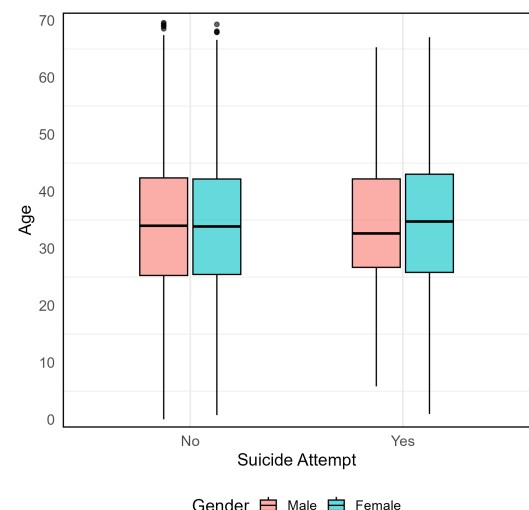

Figure A1: EHR data: age distribution by suicide-attempt status and gender.

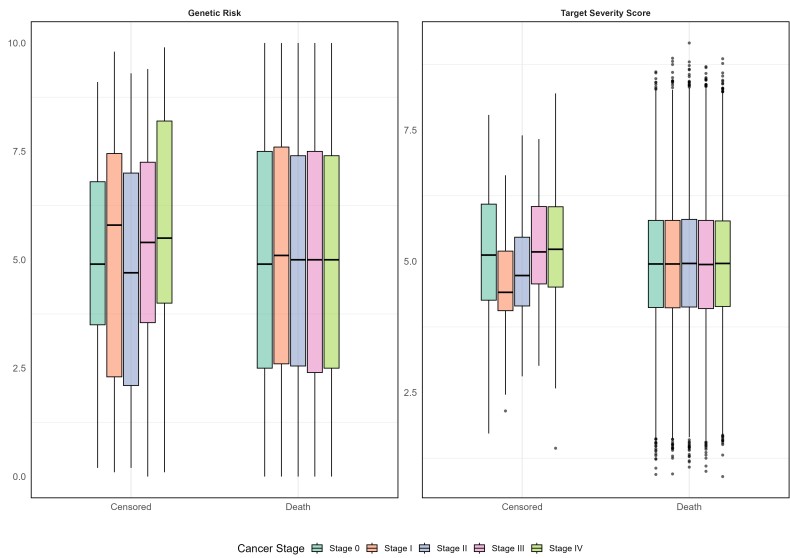

Figure A2: Machine-learning survival data: genetic-risk and target-severity score distributions by death status and cancer stage.

did not attempt suicide, yielding a censoring rate of $91.06\%$. The censoring time $C$ is the number of days from study start to end.

**Machine-learning survival data.** The cancer dataset contains $50,000$ subjects and $14$ features, including survival years, year of diagnosis, demographics (age, gender, country/region), clinical characteristics (cancer type and stage 0–IV), a composite severity score, risk factors (e.g., genetic predisposition, air pollution, alcohol use, smoking, obesity), treatment cost (USD), and survival outcomes. Figure A2 displays genetic-risk and target-severity score distributions among deaths, stratified by stage. Only $227$ individuals remain alive at study end (censored), corresponding to a $0.454\%$ censoring rate. We focus on *genetic risk* and *target severity score* as covariates. Here, $T$ is survival time in years; $\delta = 1$ if death occurred and 0 otherwise; $C$ is time from diagnosis to censoring (e.g., study end). The dataset is publicly available at Kaggle (https://www.kaggle.com/datasets/zahidmughal2343/global-cancer-patients-2015-2024).

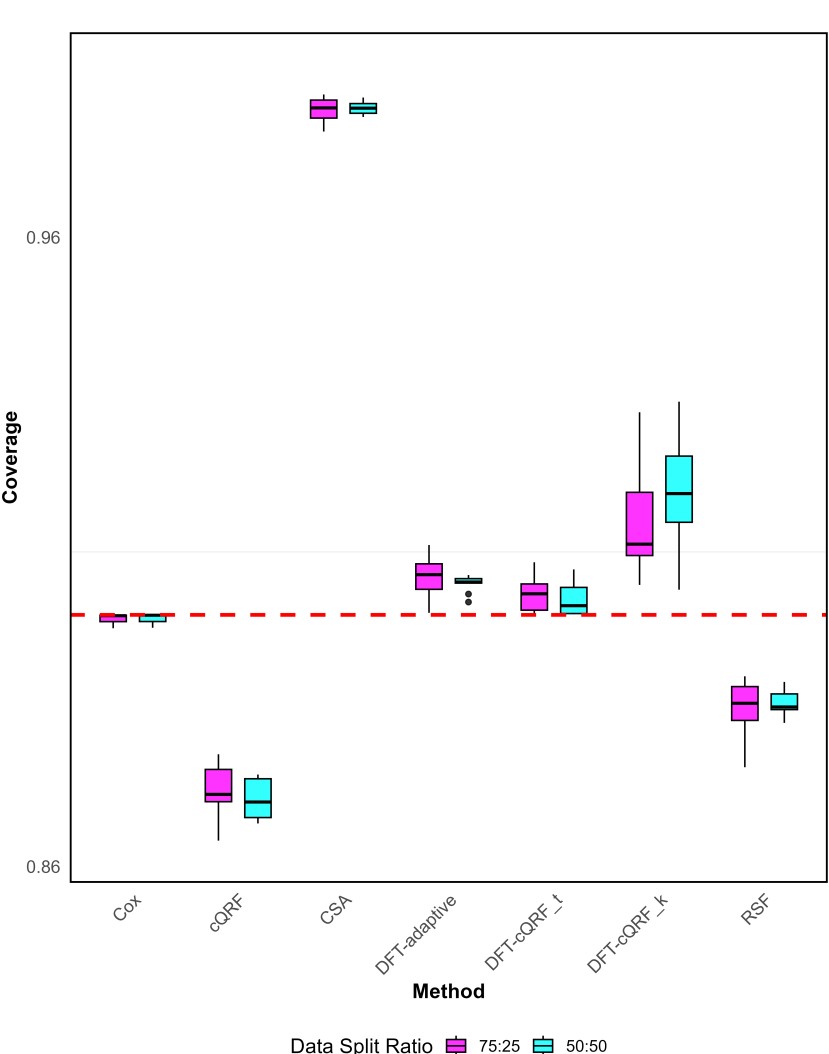

Figure A3: Analysis of the the machine-learning survival dataset: empirical coverage of all methods on .

SYNTHETIC RESULTS

Additional figures and tables for the synthetic results; all are described in Section 4

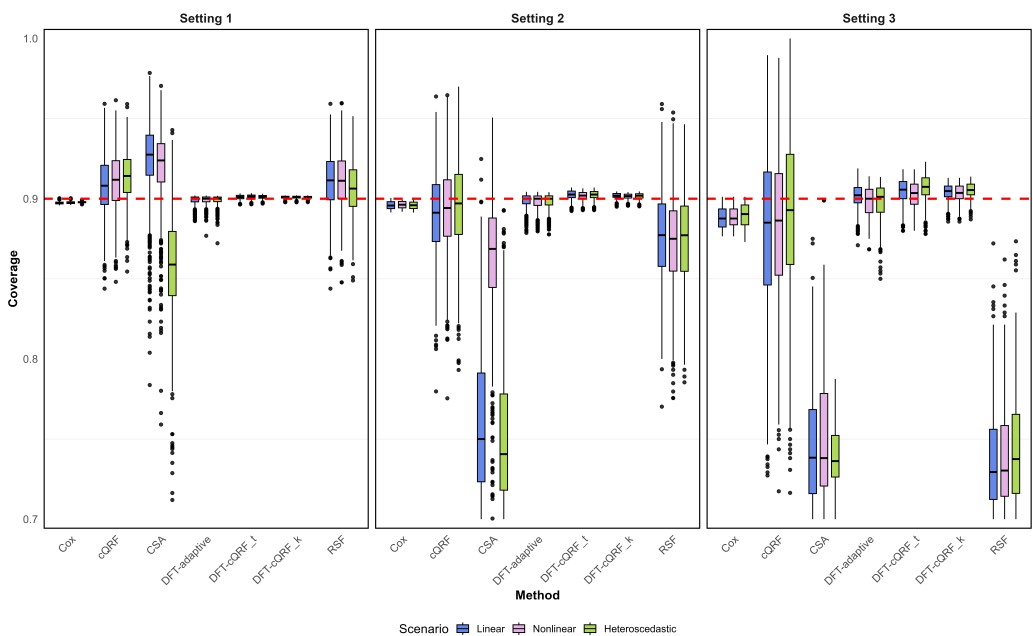

Figure A4: Synthetic experiment: empirical coverage with multiple covariates under linear, nonlinear, and heteroscedastic relationships and different censoring proportions.

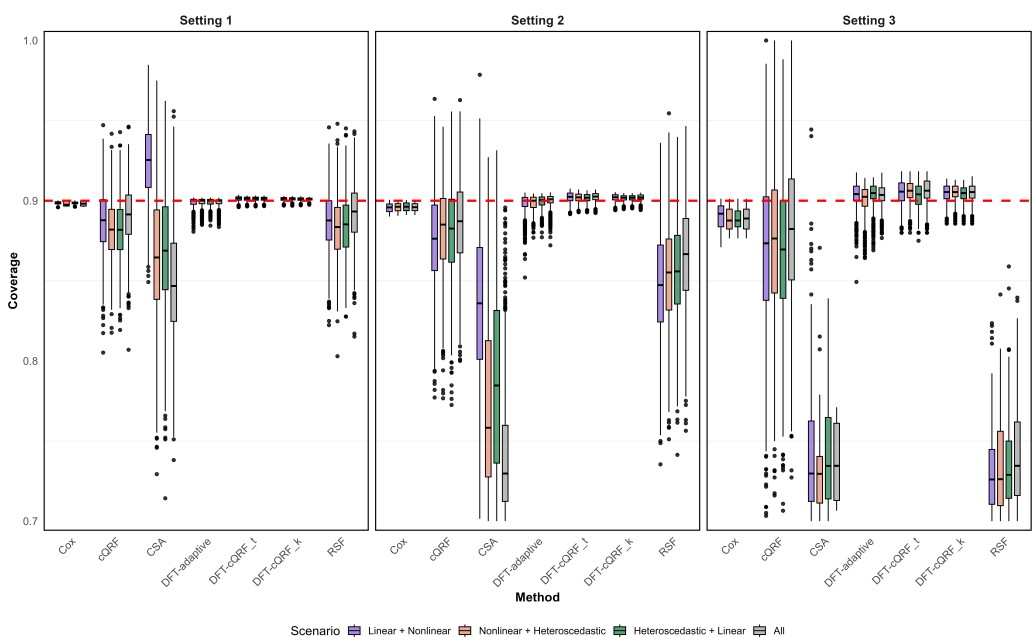

Figure A5: Synthetic experiment: empirical coverage under mixed covariate–outcome relationships and different censoring proportions.

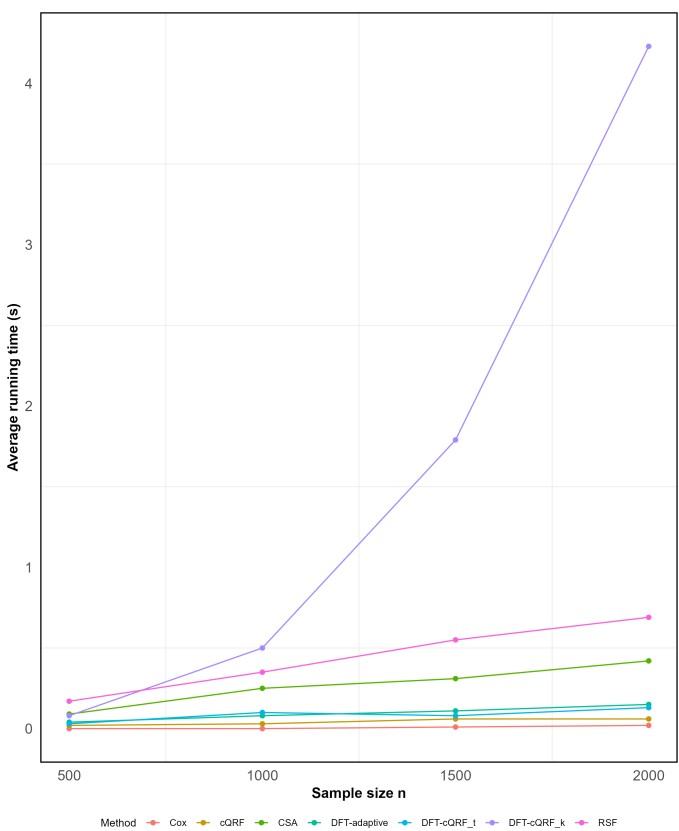

Figure A6: Synthetic experiment: running times (seconds) for the linear scenario in setting 2 with CR = 0.3 (windows 10 x64; intel core i7-12700h (14c/20t) using system.time() function).

Table A1: Synthetic experiment: average estimation error with standard deviation in setting 1 with CR = 0.

| Scenario | $n$ | Cox | CSA | cQRF | DFT-cQRF_t | DFT-cQRF_k | DFT-adaptive | RSF |
|---|---|---|---|---|---|---|---|---|
| Linear | 500 | $1.255 \pm 0.415$ | $0.061 \pm 0.009$ | $0.763 \pm 0.090$ | $\mathbf{0.054 \pm 0.002}$ | $0.057 \pm 0.012$ | $0.759 \pm 0.108$ | $0.883 \pm 0.183$ |
| | 1000 | $1.220 \pm 0.250$ | $0.056 \pm 0.002$ | $0.790 \pm 0.080$ | $\mathbf{0.051 \pm 0.006}$ | $0.053 \pm 0.005$ | $0.817 \pm 0.139$ | $0.949 \pm 0.127$ |
| | 1500 | $1.193 \pm 0.155$ | $0.060 \pm 0.006$ | $0.905 \pm 0.118$ | $\mathbf{0.052 \pm 0.004}$ | $0.058 \pm 0.005$ | $0.934 \pm 0.115$ | $1.017 \pm 0.103$ |
| | 2000 | $1.257 \pm 0.114$ | $0.065 \pm 0.007$ | $1.144 \pm 0.075$ | $\mathbf{0.062 \pm 0.008}$ | $0.064 \pm 0.008$ | $1.121 \pm 0.101$ | $1.115 \pm 0.077$ |
| Nonlinear | 500 | $1.196 \pm 0.143$ | $0.065 \pm 0.008$ | $0.999 \pm 0.159$ | $\mathbf{0.061 \pm 0.006}$ | $0.070 \pm 0.008$ | $1.088 \pm 0.129$ | $1.128 \pm 0.117$ |
| | 1000 | $1.328 \pm 0.109$ | $0.064 \pm 0.005$ | $1.131 \pm 0.078$ | $\mathbf{0.063 \pm 0.004}$ | $0.059 \pm 0.006$ | $1.180 \pm 0.055$ | $1.111 \pm 0.076$ |
| | 1500 | $1.290 \pm 0.135$ | $0.065 \pm 0.009$ | $1.165 \pm 0.131$ | $\mathbf{0.058 \pm 0.007}$ | $0.066 \pm 0.007$ | $1.083 \pm 0.070$ | $1.136 \pm 0.127$ |
| | 2000 | $1.335 \pm 0.090$ | $0.068 \pm 0.004$ | $1.182 \pm 0.063$ | $\mathbf{0.067 \pm 0.004}$ | $0.070 \pm 0.010$ | $1.111 \pm 0.062$ | $1.152 \pm 0.067$ |
| Heteroscedastic | 500 | $1.375 \pm 0.099$ | $0.063 \pm 0.007$ | $1.113 \pm 0.140$ | $\mathbf{0.058 \pm 0.009}$ | $0.060 \pm 0.019$ | $1.127 \pm 0.181$ | $1.127 \pm 0.142$ |
| | 1000 | $1.318 \pm 0.136$ | $0.070 \pm 0.007$ | $1.229 \pm 0.092$ | $\mathbf{0.072 \pm 0.013}$ | $0.077 \pm 0.011$ | $1.218 \pm 0.073$ | $1.199 \pm 0.083$ |
| | 1500 | $1.309 \pm 0.084$ | $0.073 \pm 0.007$ | $1.268 \pm 0.078$ | $\mathbf{0.071 \pm 0.007}$ | $0.074 \pm 0.005$ | $1.260 \pm 0.105$ | $1.236 \pm 0.074$ |
| | 2000 | $1.341 \pm 0.106$ | $0.068 \pm 0.006$ | $1.225 \pm 0.063$ | $\mathbf{0.064 \pm 0.004}$ | $0.070 \pm 0.007$ | $1.180 \pm 0.077$ | $1.198 \pm 0.064$ |

Table A2: Synthetic experiment: average estimation error with standard deviation in setting 3 with CR = 0.7.

| Scenario | $n$ | Cox | CSA | cQRF | DFT-cQRF_t | DFT-cQRF_k | DFT-adaptive | RSF |
|---|---|---|---|---|---|---|---|---|
| Linear | 500 | $1.337 \pm 0.202$ | $0.062 \pm 0.012$ | $1.138 \pm 0.135$ | $\mathbf{0.057 \pm 0.014}$ | $0.061 \pm 0.014$ | $1.056 \pm 0.126$ | $1.126 \pm 0.128$ |
| | 1000 | $1.218 \pm 0.087$ | $0.065 \pm 0.007$ | $1.150 \pm 0.091$ | $\mathbf{0.065 \pm 0.010}$ | $0.066 \pm 0.009$ | $1.102 \pm 0.107$ | $1.126 \pm 0.086$ |
| | 1500 | $1.305 \pm 0.123$ | $0.068 \pm 0.005$ | $1.198 \pm 0.062$ | $\mathbf{0.065 \pm 0.008}$ | $0.066 \pm 0.008$ | $1.187 \pm 0.063$ | $1.171 \pm 0.058$ |
| | 2000 | $1.257 \pm 0.114$ | $0.065 \pm 0.007$ | $1.144 \pm 0.075$ | $\mathbf{0.062 \pm 0.008}$ | $0.064 \pm 0.008$ | $1.121 \pm 0.101$ | $1.115 \pm 0.077$ |
| Nonlinear | 500 | $1.196 \pm 0.143$ | $0.065 \pm 0.008$ | $1.139 \pm 0.114$ | $\mathbf{0.061 \pm 0.006}$ | $0.070 \pm 0.008$ | $1.088 \pm 0.129$ | $1.128 \pm 0.117$ |
| | 1000 | $1.328 \pm 0.109$ | $0.064 \pm 0.005$ | $1.131 \pm 0.078$ | $\mathbf{0.063 \pm 0.004}$ | $0.059 \pm 0.006$ | $1.180 \pm 0.055$ | $1.111 \pm 0.076$ |
| | 1500 | $1.290 \pm 0.135$ | $0.065 \pm 0.009$ | $1.165 \pm 0.131$ | $\mathbf{0.058 \pm 0.007}$ | $0.066 \pm 0.007$ | $1.083 \pm 0.070$ | $1.136 \pm 0.127$ |
| | 2000 | $1.335 \pm 0.090$ | $0.068 \pm 0.004$ | $1.182 \pm 0.063$ | $\mathbf{0.067 \pm 0.004}$ | $0.070 \pm 0.010$ | $1.111 \pm 0.062$ | $1.152 \pm 0.067$ |
| Heteroscedastic | 500 | $1.375 \pm 0.099$ | $0.063 \pm 0.010$ | $1.151 \pm 0.129$ | $\mathbf{0.058 \pm 0.009}$ | $0.060 \pm 0.019$ | $1.127 \pm 0.181$ | $1.127 \pm 0.142$ |
| | 1000 | $1.318 \pm 0.136$ | $0.070 \pm 0.007$ | $1.229 \pm 0.092$ | $\mathbf{0.072 \pm 0.013}$ | $0.077 \pm 0.011$ | $1.218 \pm 0.073$ | $1.199 \pm 0.083$ |
| | 1500 | $1.309 \pm 0.084$ | $0.073 \pm 0.007$ | $1.268 \pm 0.078$ | $\mathbf{0.071 \pm 0.007}$ | $0.074 \pm 0.005$ | $1.260 \pm 0.105$ | $1.236 \pm 0.074$ |
| | 2000 | $1.341 \pm 0.106$ | $0.068 \pm 0.006$ | $1.225 \pm 0.063$ | $\mathbf{0.064 \pm 0.004}$ | $0.070 \pm 0.007$ | $1.180 \pm 0.077$ | $1.198 \pm 0.064$ |

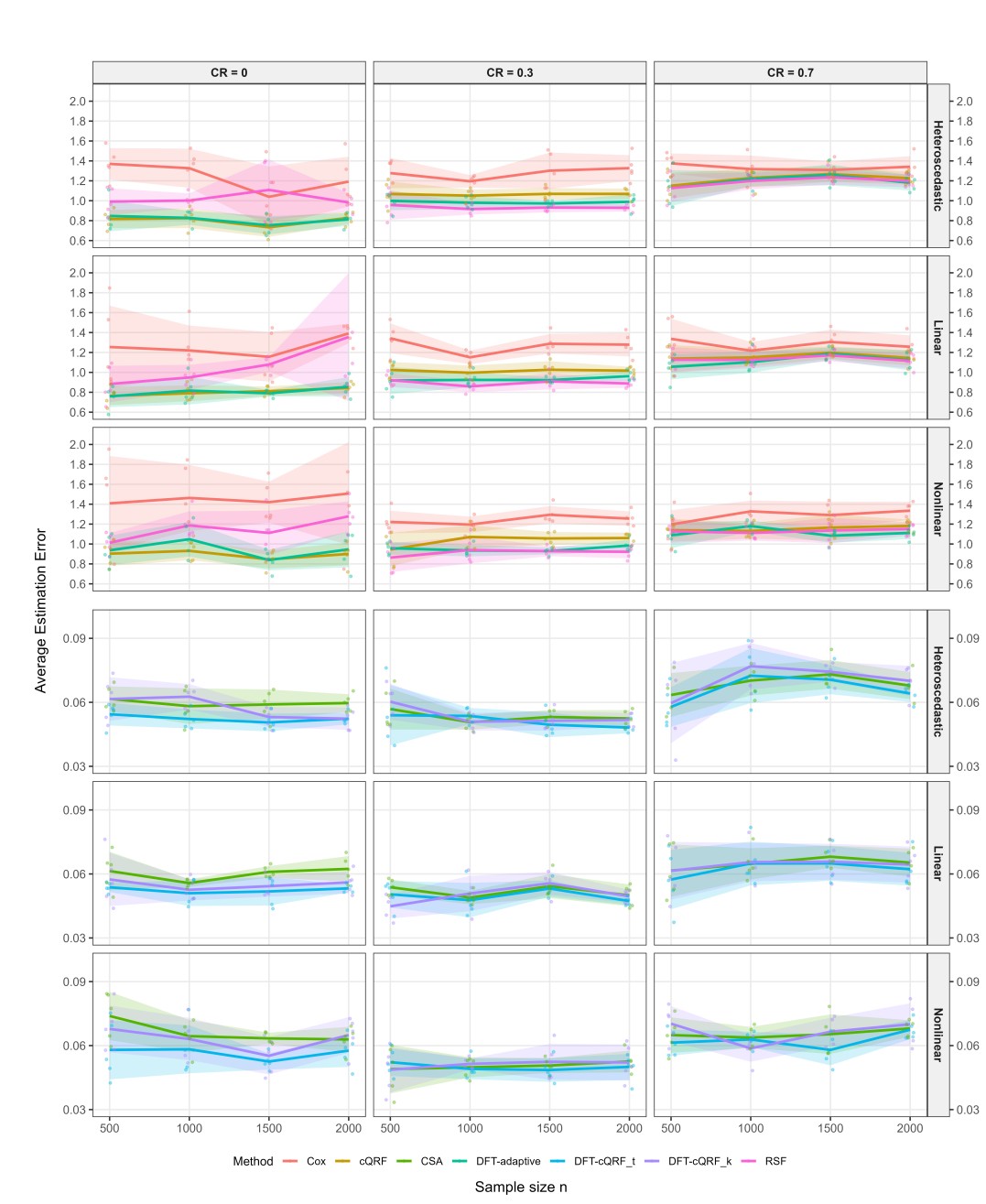

Figure A7: Synthetic experiment: average estimation error across settings and scenarios.

