# OpenReview forum: "Distribution-Free Lower Predictive Bounds for Right-Censored Time-to-Event Data via Hybrid Quantile Learning and DFT-Adaptive Conformal Calibration"
_ICLR.cc/2026/Conference — ICLR 2026 Conference Withdrawn Submission_

### Official Review · Reviewer_hZcz · 2025-10-14

**Soundness:** 3
**Presentation:** 1
**Contribution:** 2
**Rating:** 2
**Confidence:** 4

**Summary:**

The paper introduces DFT–cQRF, a hybrid, model-agnostic framework that combines (i) flexible conditional quantile machine learning tools to estimate the conditional quantile of the event time under right censoring; (ii) conformal inference framework with a data-filtered threshold to provide individualized predictions for the lower bound of the survival time.

Disclaimer: I used LLMs to polish some of my reviews in terms of grammars, but not all reviews.

**Strengths:**

Literature review is comprehensive. Experiment is extensive, both real-data and simulated data results are presented, which are helpful and showcase the practicality of the methods. Simulation results on coverage rate are convenicing to support the method.

**Weaknesses:**

The paper has a number of weaknesses regarding the fundamental novelty and presentations. For presentation issues such as figure quality and clarity, language polishing etc. (see more details below), I believe they are quite straightforward to be addressed and improved, so I am not too worry about this part. However, the main issue is about the technical novelty (see Major comments). If the author(s) believe this can be well-addressed (as well as all my other comments), I am certainly open to change my assessment and score. I therefore still encourage the author(s) to consider a rebuttal to my comments.

**Major comments**

While I acknowledge that the paper presents some innovative aspects, its contribution relative to existing work remains somewhat limited. In particular, the authors themselves reference **"Farina, R., Tchetgen, E. J. T., and Kuchibhotla, A. K. (2024). Doubly robust and efficient calibration of prediction sets for censored time-to-event outcomes,"** which already develops a robust and efficiency-oriented conformal calibration framework for censored data. The current paper’s advancement beyond that line of work is not sufficiently clarified, and the distinction between the proposed hybrid quantile-based approach and the doubly robust conformal methods should be more explicitly articulated.

Specifically, Farina et al. also proposed an IPCW estimator. Therefore, while your method has some algorithm novelty, is it fundamentally the same as their IPCW estimator but only differs in implementation details? This is an important point to clarify otherwise the paper lacks good novelty. If you believe your method has some **fundamental** and **technical** aspects that differ from their estimator, please specify in details.

Farina et al. further proposed a doubly robust estimator (AIPWC). If I understand correctly, they do not require all nuisance functions such as censoring and survival to all be correctly specified to ensure the marginal coverage of the conformal intervals, which gives more chance for valid predictions since the nuisance $G(t\mid x)$ in your method is subject to misspecification.

At the same time, I wonder if there is anyway to compare your methods to theirs? They have some simulation code here: https://github.com/rebyfa98/DR_conformal_censored If it is not too hard to understand and implement their code, it would be nice to see some supplemental results.

The key of this paper is Theorems 1 and 2, but there are too many regularity conditions listed (those in Appendix A). It is understandable that for theoretical results, they need to be assumed, but how possible they always hold in practice? Please have some plain and brief language explanations/summary for some important and somewhat more abstract conditions (such as A3, K2, C4) in the main text. And would the set of these regularity conditions be much more compared to those in Farina et al.?

**Other comments**

- The first sentence of the paper: Time-to-event prediction is central to decision-making. This is an overly strong statement. There are lots of different types of variables, not just time-to-event. Suggest reducing the tone.

- Introduction, line 45 "... Conformal prediction provides model-agnostic procedures with minimal assumptions..." Minimal is a strong word and subjective. Suggest revising to mild or something.

- Line 45: Why exchangeability of data is a minimal assumption? What does it mean? It is pretty much vague here. In my understanding, exchangeability can be strong, e.g., across heterogenous groups, even conditional on some predictors is still not enough to make inference on the interested variable because of label shift. This is something often not testable in practice. I don't see a point why you want to mention it here, or maybe the tone needs to be revised, such as "the goal of conformal inference is under some exchangeability assumptions, one can construct individualized prediction intervals..."

- Avoid double (( )) shows, such as "(e.g., Vovk et al. (2005))"...

- You reviewed a lot of papers and methods for conformal inference. However, the introduction should be more focused on the knowledge gap. You did not specified why your method is unique and different than these existing methods. For example, in the paragraph starts with line 52, I only see you reviewed efforts for conformal survival analysis, but what are things they did not address but you addressed?

- Some papers are not reviewed yet popular and important:

  - Lei, Lihua, and Emmanuel J. Candès. "Conformal inference of counterfactuals and individual treatment effects." Journal of the Royal Statistical Society Series B: Statistical Methodology 83.5 (2021): 911-938.

  - Yang, Yachong, Arun Kumar Kuchibhotla, and Eric Tchetgen Tchetgen. "Doubly robust calibration of prediction sets under covariate shift." Journal of the Royal Statistical Society Series B: Statistical Methodology 86.4 (2024): 943-965.

- Typo: naively -> na\\"ively (\\" is used in latex).

As a result, in my opinion, the entire Section 1 needs to be polished and shaped better based on my comments above.

- It would be helpful to clarify in the early of the method section that whether your method is designed for discrete or continuous time or both. This is an important concern for survival analysis.

- Line 120: you mentioned you use lower cases to denote the realized values, yet $\delta$ does not have a non-realized version. Should you use $\Delta$ in the iid data copy and $\delta$ as its realization?

- Define once and only once any abbreviations. There are multiple definitions of IPCW.

- **All figures are not visible.** Figure 1 is nice for the flow, but because of the vague texts and quality, people may not really wants to read the details. The same issue affects the other figures, possibly because the authors reduced the figure heights in an attempt to save space. I am not sure if figures are allowed to uploaded as links during the rebuttal for me to check, if it is not allowed I am okay you do this later. Nevertheless, I strongly recommend you to make the figure quality better in the next version.

- In all figures, the texts and legends are way too small to read. As I mentioned above, some figures are great, but if they have low quality people will ignore their values...

- The smallest sample size of the simulated data starts from 500, but it would also be interesting to see results under very small sample sizes, such as 100. This is common in many real-world clinical trials for survival endpoints. Also, honestly 500, 1000, 1500, 2000 is not a very good choice for the sample size grid, since they are too close. I would rather do 100, 500, 1000, 5000... This allows seeing whether the coverage becomes more precise proportional to the sample size. If n = 5000 can take too long to run during the rebuttal, I understand, but I strongly recommend you consider it to see the changes to the sample sizes more clearly.

- Besides the coverage, I think it is also important to report the values of the lower prediction bound of event time. This is because if the lower prediction bound is too small or close to 0, it won't be informative for practice. For simulation, you can think about reporting the mean (sd) of the lower bound over replicates. For your real data, you can report the same or pick the results of one or two participants to report.

- While the results of coverage of the proposed methods are best among competing methods, they are often above the nominal coverage lines, indicating conservative predictions. It is worthy to provide some explanations of why such phenomenon is frequently observed.

- Section 5 is too short. Please elaborate and propose how would you extend your methods in other scenarios in more details. You should also cite more papers to support these extensions, especially why these extensions are worthy to consider in practice. Why two-sided conformal intervals present more challenges?

- Discussion on involving time-varying covariates are also of importance for future research. In survival analysis, baseline information are not always the most predictive.

- Some references are not written well enough, e.g., "Farina, R., Tchetgen, E. J. T., and Kuchibhotla, A. K. (2024). Doubly robust and efficient calibration of prediction sets for censored time-to-event outcomes. arXiv preprint. Preprint; please update arXiv ID" shown in your References at the end of main paper. Here "please update arXiv ID" looks like an error. Please check **all references** are correctly cited.

**Questions:**

No much to add here. See weaknesses above.

---

### Official Review · Reviewer_8f4z · 2025-10-31

**Soundness:** 1
**Presentation:** 2
**Contribution:** 1
**Rating:** 2
**Confidence:** 4

**Summary:**

The paper focuses on the problem of constructing lower prediction bounds for right-censored time-to-event data, where the right censoring may depend on covariates. They propose an algorithm to first estimate the maximum support of censoring conditional on covariates and restrict their attention to subjects whose event time is observed and is less than this threshold, and construct a lower prediction bound for this sub-population. In particular, they propose incorporating IPCW weights to quantile random forest for estimating censoring weights and use a kernel-based method to estimate the maximum support of censoring conditional on covariates. They provide theory for the asymptotic properties of their proposed estimators and coverage results for their constructed lower prediction bounds. The performance of the proposed approach is illustrated with simulated and real-world data.

**Strengths:**

The pipeline diagram and algorithm are helpful in illustrating the complex procedures of the approach.

**Weaknesses:**

There seems to be fundamental issues with their approach. They consider the threshold $\tau(x)$ and only restrict to subjects that are uncensored and with event time less than the threshold when computing the quantile for the quantile residuals. It is known in the survival literature that the upper tail distribution of the event time is not identifiable after the maximum follow up time, but this should not be a problem for constructing a lower prediction bound (it may need to be handled for upper prediction bound). They seem to misunderstand and role of the cut-off threshold in the Candes et al. (2023) and Gui et al. (2024) papers. The cut-off in their paper is not used for restricting the target population of interest to a smaller subgroup, but rather for applying the result that the LPB based on the censored event time is naturally a valid LPB for the event time of interest.

In step 3 of the method, the authors only consider the residual for subjects that are uncensored and with the event time lower than $\hat\tau(x_i)$; then in steps 4 and 5, the $1-\alpha$ quantile of those residuals is used to construct the LPB. Their coverage guarantees are for this restricted population. However, the goal in practice is to construct prediction sets for time-to-event outcomes that have guaranteed coverage in the entire population, not only among the uncensored ones; this is where the challenge of censoring comes in. With interest only in the uncensored population, conformal prediction approaches for continuous outcomes can be directly applied.

The introduction and literature review also need to be polished. More clarity in positioning the paper in the broader literature and how the paper compares with existing approaches.

**Questions:**

1. What censoring assumption is needed for the proposed method? Can you be more specific on what you mean by censoring varies strongly with covariates, etc.?

2. The paper focuses on using IPCW weighted quantile random forest. Can the method be applied with other machine learning approaches? How does your approach compare with applying Survival Random Forest to estimate the censoring distribution? Is cQRF just a different algorithm to estimate the censoring survival probability?

3. For equation (1), the value of $\hat\tau(x)$ seems to depend on what kernel is used. For example, if the Gaussian kernel is used, would $\hat\tau(x)$ will be the maximum censoring times for all $x$?

---

### Official Review · Reviewer_ogaY · 2025-11-01

**Soundness:** 3
**Presentation:** 1
**Contribution:** 2
**Rating:** 4
**Confidence:** 3

**Summary:**

The paper aims to produce distribution-free lower bounds for time-to-event under right censoring, with guarantees that hold in finite samples conditional on the test case being observable and with an asymptotically vanishing correction so the bound converges to the true conditional quantile.

The method pairs a censoring-aware quantile learner (cQRF with IPCW) with a covariate-adaptive censoring horizon, then performs split-conformal calibration only on observable points. Among all constant shifts that achieve the target coverage, the conformal shift is minimal.

Extensive empirical validation on two real-world datasets and synthetic settings verifies the validity and tightness.

**Strengths:**

1. A sound split-conformal framework that guarantees finite-sample coverage for test points that are uncensored and within the learned horizon, and is asymptotically sharp.

2. Extensive empirical validation: Extensive empirical validation on two real-world datasets (EHR with 91.06% censoring; cancer with 0.454% censoring) and comprehensive synthetic settings spanning linear, nonlinear, and heteroscedastic relationships under different censoring. Empirical results show coverage close to the nominal level with lower variability.

**Weaknesses:**

1. Data efficiency: Calibration uses only “observable” points. Under heavy censoring the effective calibration size can be small or zero; the lower bound becomes vacuous.

2. Moderate novelty: The method is incremental over adaptive-cutoff conformal survival: it adds a covariate-dependent censoring horizon and a censoring-aware quantile learner combined with a constant split-conformal shift.

3. Presentation: Several box plots are very small with some distorted aspect ratios, and section formatting (e.g. methodology as a paragraph) can be hard to scan.

**Questions:**

See weakness.

---

### Official Review · Reviewer_C4hL · 2025-11-09

**Soundness:** 1
**Presentation:** 3
**Contribution:** 1
**Rating:** 2
**Confidence:** 4

**Summary:**

The paper proposes a conformal-survival–style method for constructing **distribution-free lower prediction bounds** for time-to-event outcomes, claiming to handle **covariate-dependent (heterogeneous) censoring**. Methodologically, it extends prior CSA/adaptive-CSA ideas by modeling a conditional censoring horizon $G(t \mid X)$ and using it to define conformity scores and calibrated lower bounds. The authors evaluate the approach on synthetic data and two “real-world” datasets (EHR and a Kaggle survival dataset), and report improved empirical coverage and interval tightness compared to existing conformal-survival baselines.

**Strengths:**

N/A

**Weaknesses:**

1. A major (and completely unacknowledged) limitation is that the proposed method is only applicable under Type-I censoring, i.e., settings where the censoring time is known for *every* individual, regardless of whether the event is observed. In that case, the only censoring mechanism is administrative censoring at the end of the study. This is a very special design that rarely holds in real medical applications, where censoring typically arises from a mixture of causes: study termination, patient dropout, loss to follow-up, competing risks, etc.

   In the real-world data analysis (EHR and Kaggle datasets), the authors simply set the censoring time of **every** patient to the end of the study. For patients who actually experience the event (death, suicide attempt, etc.), these “censoring times” are not real censoring times at all, but rather the authors’ imagined counterfactual “time they would have been censored if the only censoring mechanism were administrative.” In reality, once a patient has died, we never observe whether they would have dropped out earlier, experienced a competing event, or remained under follow-up.

   As a result, these so-called “real-world” datasets are in fact **semi-synthetic**: they contain partially real labels (event times) and partially synthetic censoring times. Given that some prior conformal survival work (CSA, adaptive CSA) has also operated in this very restricted Type-I setting, it is not unreasonable to study this regime. However, the paper should *clearly acknowledge* that the method and experiments are tailored to this artificial censoring design and are not directly applicable to more realistic censoring mechanisms.

2. A second, and even more serious, conceptual issue is the **incompatibility between Type-I censoring and the claimed “covariate-dependent” (heterogeneous) censoring**. Under Type-I censoring, everyone is censored at the study end (or at a fixed administrative time), so the only factor that can influence the censoring time is when a subject enrolled in the study. Equivalently, the censoring time $C$ does **not** depend on patient covariates $X$. In fact, in the standard survival analysis literature, Type-I censoring is the canonical example of random censoring with $T \perp C$.

   If the paper truly assumes Type-I censoring (as is effectively enforced in the EHR and Kaggle analyses by setting all censoring times to the study end), then no patient covariates should affect the censoring distribution $G(t \mid X)$. In that regime, modeling censoring as covariate-dependent and fitting a complex conditional censoring model can only introduce variance/overfitting, not genuine “heterogeneous censoring” structure or improved accuracy. This criticism applies both to the current work and to the prior adaptive CSA paper (Gui 2024): in a pure Type-I design, the advertised benefit of “adapting to heterogeneous censoring” is conceptually inconsistent with the underlying censoring mechanism.

3. Synthetic experiments do not contain heterogeneous censoring. This point relates to point 2, the synthetic experiments use censoring times generated as $C \sim \mathrm{Unif}(0,\lambda)$ with a *constant* $\lambda$. This implies that censoring is independent of covariates, so there is **no heterogeneous censoring** in any of the synthetic settings. Consequently, it is unclear why methods that explicitly “account for heterogeneous censoring” should be expected to outperform simpler approaches such as CSA in these simulations.

   The same issue appears in the two “real-world” datasets. The paper fits conditional censoring models using covariates such as age and gender to estimate $G(t \mid X)$, but the constructed censoring times are just the study end for everyone and therefore, by construction, *do not* depend on these covariates. Again, the paper claims to handle covariate-dependent censoring, but all of its experiments take place in regimes where censoring is effectively independent of $X$.

4. The calibration step is also problematic. The method only performs conformal calibration on the subset of the calibration set with finite conformity scores (i.e., subjects whose scores are not set to $\infty$). In practice, this can remove a substantial fraction of calibration points, especially in heavily censored settings, leaving a small effective calibration sample. This design almost inevitably pushes the procedure towards conservative intervals (since calibration is being done on a restricted, cherry-picked subset), and it is unclear how coverage behaves outside this subset. Given the much more fundamental issues in the censoring assumptions described above, I consider this a secondary problem, but it further weakens the overall methodology.

**Questions:**

Please see above.

---

### Note · Authors · 2025-12-02

I have read and agree with the venue's withdrawal policy on behalf of myself and my co-authors.